# Computationally Efficient Horizon-Free Reinforcement Learning for Linear Mixture MDPs

**Dongruo Zhou**
Department of Computer Science
University of California, Los Angeles
Los Angeles, CA 90095
drzhou@cs.ucla.edu

**Quanquan Gu**
Department of Computer Science
University of California, Los Angeles
Los Angeles, CA 90095
qgu@cs.ucla.edu

## Abstract

Recent studies have shown that episodic reinforcement learning (RL) is not more difficult than contextual bandits, even with a long planning horizon and unknown state transitions. However, these results are limited to either tabular Markov decision processes (MDPs) or computationally inefficient algorithms for linear mixture MDPs. In this paper, we propose the first computationally efficient horizon-free algorithm for linear mixture MDPs, which achieves the optimal $\widetilde{O}(d\sqrt{K} + d^2)$ regret up to logarithmic factors. Our algorithm adapts a weighted least square estimator for the unknown transitional dynamic, where the weight is both *variance-aware* and *uncertainty-aware*. When applying our weighted least square estimator to heterogeneous linear bandits, we can obtain an $\widetilde{O}(d\sqrt{\sum_{k=1}^{K} \sigma_k^2} + d)$ regret in the first $K$ rounds, where $d$ is the dimension of the context and $\sigma_k^2$ is the variance of the reward in the $k$-th round. This also improves upon the best-known algorithms in this setting when $\sigma_k^2$'s are known.

## 1  Introduction

How to design efficient algorithms is a central problem for reinforcement learning (RL). Here, the *efficiency* includes both *statistical efficiency*, which requires the RL algorithm enjoy a low regret/polynomial sample complexity for finding the near-optimal policy, and *computational efficiency*, which expects the RL algorithm have polynomial running time. When restricting to episodic RL with total reward upper bounded by $1$[1], a longstanding question is whether episodic RL is statistically and computationally more difficult than contextual bandits (Jiang and Agarwal, 2018), since episodic RL can be seen as an extension of contextual bandits to have a long planning horizon and unknown state transition. For tabular RL, this questions has been fully resolved by a line of works (Wang et al., 2020; Zhang et al., 2021b; Li et al., 2022; Zhang et al., 2022), which propose various horizon-free algorithms. Here we say an algorithm is horizon-free if its regret/sample complexity has at most a polylogarithmic dependence on the planning horizon $H$. In particular, (Zhang et al., 2021b) proposed the first computationally efficient algorithm for tabular RL whose regret enjoys a polylogarithmic dependence on the planning horizon, and (Zhang et al., 2022) further removed the polylogarithmic dependence on the planning horizon.

For RL with *function approximation* to deal with large state space, Zhang et al. (2021c); Kim et al. (2021) have made some progress towards horizon-free RL for a class of MDPs called *linear mixture MDPs* (Jia et al., 2020; Ayoub et al., 2020; Zhou et al., 2021a), whose transition dynamic can be represented as a linear combination of $d$ basic transition models. More specifically, Zhang et al.

---

[1]See Assumption 3.1 for a detailed description.

(2021c) proposed a VARLin algorithm for linear mixture MDPs with an $\widetilde{O}(d^{4.5}\sqrt{K} + d^9)$ regret for the first $K$ episodes, and Kim et al. (2021) proposed a VARLin2 algorithm with an improved regret $\widetilde{O}(d\sqrt{K} + d^2)$. However, neither algorithm is computationally efficient, because both of them need to work with nonconvex confidence sets and do not provide a polynomial-time algorithm to solve the maximization problem over these sets.

So the following question remains open:

*Can we design computationally efficient horizon-free RL algorithms when function approximation is employed?*

In this paper, we answer the above question affirmatively for linear function approximation by proposing the first computationally efficient horizon-free RL algorithm for linear mixture MDPs. Our contributions are summarized as follows.

- As a warm-up, we consider the heterogeneous linear bandits where the variances of rewards in each round are different. Such a setting can be regarded as a special case of linear mixture MDPs. We propose a computationally efficient algorithm WeightedOFUL$^+$ and prove that in the first $K$-rounds, the regret of WeightedOFUL$^+$ is $\widetilde{O}(d\sqrt{\sum_{k=1}^{K} \sigma_k^2} + dR + d)$, where $\sigma_k^2$ is the variance of the reward in the $k$-th round and $R$ is the magnitude of the reward noise. Our regret is *variance-aware*, i.e., it only depends on the summation of variances and does not have a $\sqrt{K}$ term. This directly improves the $\widetilde{O}(d\sqrt{\sum_{k=1}^{K} \sigma_k^2} + \sqrt{dK} + d)$ regret achieved by Zhou et al. (2021a).

- For linear mixture MDPs, when the total reward for each episode is upper bounded by 1, we propose a HF-UCRL-VTR$^+$ algorithm and show that it has an $\widetilde{O}(d\sqrt{K} + d^2)$ regret for the first $K$ episodes, where $d$ is the number of basis transition dynamic. Our HF-UCRL-VTR$^+$ is *computationally efficient*, *horizon-free* and *near-optimal*, as it matches the regret lower bound proved in our paper up to logarithmic factors. Our regret is strictly better than the regret attained in previous works (Zhou et al., 2021a; Zhang et al., 2021c; Kim et al., 2021).

- At the core of both WeightedOFUL$^+$ and HF-UCRL-VTR$^+$ is a carefully designed *weighted linear regression* estimator, whose weight is both *variance-aware* and *uncertainty-aware*, in contrast to previous weighted linear regression estimator that is only *variance-aware* (Zhou et al., 2021a). For linear mixture MDPs, we further propose a HOME that constructs the variance-uncertainty-aware weights for high-order moments of the value function, which is pivotal to obtain a horizon-free regret for linear mixture MDPs.

For a better comparison between our results and previous results, we summarize these results in Table 1. It is evident that our results improve upon all previous results in the respective settings[2].

**Notation.** We use lower case letters to denote scalars, and use lower and upper case bold face letters to denote vectors and matrices respectively. We denote by $[n]$ the set $\{1, \ldots, n\}$, by $\overline{[n]}$ the set $\{0, \ldots, n-1\}$. For a vector $\mathbf{x} \in \mathbb{R}^d$ and a positive semi-definite matrix $\mathbf{\Sigma} \in \mathbb{R}^{d \times d}$, we denote by $\|\mathbf{x}\|_2$ the vector's Euclidean norm and define $\|\mathbf{x}\|_{\mathbf{\Sigma}} = \sqrt{\mathbf{x}^\top \mathbf{\Sigma} \mathbf{x}}$. For $\mathbf{x}, \mathbf{y} \in \mathbb{R}^d$, let $\mathbf{x} \odot \mathbf{y}$ be the Hadamard (componentwise) product of $\mathbf{x}$ and $\mathbf{y}$. For two positive sequences $\{a_n\}$ and $\{b_n\}$ with $n = 1, 2, \ldots$, we write $a_n = O(b_n)$ if there exists an absolute constant $C > 0$ such that $a_n \leq Cb_n$ holds for all $n \geq 1$ and write $a_n = \Omega(b_n)$ if there exists an absolute constant $C > 0$ such that $a_n \geq Cb_n$ holds for all $n \geq 1$. We use $\widetilde{O}(\cdot)$ to further hide the polylogarithmic factors. We use $\mathbb{1}\{\cdot\}$ to denote the indicator function. For $a, b \in \mathbb{R}$ satisfying $a \leq b$, we use $[x]_{[a,b]}$ to denote the truncation function $x \cdot \mathbb{1}\{a \leq x \leq b\} + a \cdot \mathbb{1}\{x < a\} + b \cdot \mathbb{1}\{x > b\}$.

## 2 Related work

In this section, we will review prior works that are most relevant to ours.

---

[2]The only exception is that for heterogeneous linear bandits, our algorithm needs to know the noise variance, while Zhang et al. (2021c); Kim et al. (2021) do not. But their algorithms are computationally inefficient. We plan to extend our algorithm to deal with unknown variance in the future work.

Table 1: Comparisons of regrets for linear bandits and linear mixture MDPs.

| Algorithm | Regret | Assumption | Computationally Efficient? |
|---|---|---|---|
| OFUL (Abbasi-Yadkori et al., 2011) | $\widetilde{O}(d\sqrt{K})$ | - | Yes |
| WeightedOFUL (Zhou et al., 2021a) | $\widetilde{O}(d\sqrt{\sum_{k=1}^{K}\sigma_k^2}+\sqrt{dK}+d)$ | Known variance | Yes |
| VOFUL (Zhang et al., 2021c) | $\widetilde{O}(d^{4.5}\sqrt{\sum_{k=1}^{K}\sigma_k^2}+d^5)$ | Unknown variance | No |
| VOFUL2 (Kim et al., 2021) | $\widetilde{O}(d^{1.5}\sqrt{\sum_{k=1}^{K}\sigma_k^2}+d^2)$ | Unknown variance | No |
| WeightedOFUL$^+$ (Theorem 4.1) | $\widetilde{O}(d\sqrt{\sum_{k=1}^{K}\sigma_k^2}+d)$ | Known variance | Yes |
| UCRL-VTR (Jia et al., 2020; Ayoub et al., 2020) | $\widetilde{O}(d\sqrt{H^3K})$ | Homogeneous, $\sum_h r_h \leq H$ | Yes |
| UCRL-VTR$^+$ (Zhou et al., 2021a) | $\widetilde{O}(\sqrt{d^2H^3+dH^4}\sqrt{K}$ $+d^2H^3+d^3H^2)$ | Inhomogeneous, $\sum_h r_h \leq H$ | Yes |
| VARLin (Zhang et al., 2021c) | $\widetilde{O}(d^{4.5}\sqrt{K}+d^9)$ | Homogeneous, $\sum_h r_h \leq 1$ | No |
| VARLin2 (Kim et al., 2021) | $\widetilde{O}(d\sqrt{K}+d^2)$ | Homogeneous, $\sum_h r_h \leq 1$ | No |
| HF-UCRL-VTR$^+$ (Theorem 5.1) | $\widetilde{O}(d\sqrt{K}+d^2)$ | Homogeneous, $\sum_h r_h \leq 1$ | Yes |
| Lower bound (Theorem 5.3) | $\Omega(d\sqrt{K})$ | - | - |

**Heterogeneous linear bandits.** Linear bandits have been studied for a long time. Most of existing works focus on the homogeneous linear bandits where the noise distributions at different round are identical (Auer, 2002; Chu et al., 2011; Li et al., 2010; Dani et al., 2008; Abbasi-Yadkori et al., 2011; Li et al., 2019a,b). Recently, a series of works focus on the heterogeneous linear bandits where the noise distribution changes over time. Lattimore et al. (2015) assumed that the noise distribution is Bernoulli and proposed an algorithm with an $\widetilde{O}(d\sqrt{K})$ regret. Kirschner and Krause (2018) assumed that the noise at $k$-th round is $\sigma_k^2$-sub-Gaussian, and they proposed a weighted ridge regression-based algorithm with an $\widetilde{O}(d\sqrt{\sum_{k=1}^{K}\sigma_k^2})$ regret. A recent line of works assume the variance of the noise at $k$-th round is bounded by $\sigma_k^2$. Under this assumption, Zhang et al. (2021c) proposed a VOFUL algorithm with an $\widetilde{O}(d^{4.5}\sqrt{\sum_{k=1}^{K}\sigma_k^2}+d^5)$ regret. Kim et al. (2021) proposed a VOFUL2 algorithm with an $\widetilde{O}(d^{1.5}\sqrt{\sum_{k=1}^{K}\sigma_k^2}+d^2)$ regret. Both VOFUL and VOFUL2 do not need to know the variance information. However, they are computationally inefficient since they need to work with nonconvex confidence sets defined by a series of second-order constraints, and do not propose a polynomial-time algorithm to solve the maximization problem over these sets. With the variance information, Zhou et al. (2021a) proposed a computationally efficient WeightedOFUL with an $\widetilde{O}(d\sqrt{\sum_{k=1}^{K}\sigma_k^2}+\sqrt{dK}+d)$ regret. Our work is under the same assumptions as Zhou et al. (2021a) and improves the regret for linear bandits.

**Horizon-free tabular RL.** RL is widely believed to be harder than contextual bandits problem due to its long planning horizon and the unknown state transitions. For tabular RL, under the assumption that the total reward obtained by any policy is upper bounded by 1, Jiang and Agarwal (2018) conjectured that any algorithm to find an $\epsilon$-optimal policy needs to have a polynomial dependence on the planning horizon $H$ in the sample complexity. Such a conjecture was firstly refuted by Wang et al. (2020) by proposing a horizon-free algorithm with an $\widetilde{O}(|\mathcal{S}|^5|\mathcal{A}|^4\epsilon^{-2}\text{polylog}(H))$ sample complexity that depends on $H$ polylogarithmically, where $\epsilon$ is the target sub-optimality of the policy, $\mathcal{S}$ is the state space and $\mathcal{A}$ is the action space. Zhang et al. (2021b) proposed a near-optimal algorithm with an improved regret $O((\sqrt{|\mathcal{S}||\mathcal{A}|K}+|\mathcal{S}|^2|\mathcal{A}|)\text{polylog}(H))$ and sample complexity. Similar

regret/sample complexity guarantees with a polylogarithmic $H$ dependence have also been established under different RL settings (Zhang et al., 2020; Ren et al., 2021; Tarbouriech et al., 2021). Recently Li et al. (2022); Zhang et al. (2022) further proposed algorithms with $H$-independent regret/sample complexity guarantees. However, all the above works are limited to tabular RL. Our work proposes an algorithm with a regret bound that depends on $H$ polylogarithmically for linear mixture MDPs, which extends these horizon-free tabular RL algorithms.

**RL with linear function approximation.** Recent years have witnessed a trend on RL with linear function approximation (e.g., Jiang et al., 2017; Dann et al., 2018; Yang and Wang, 2019a; Jin et al., 2020; Wang et al., 2019; Du et al., 2019; Sun et al., 2019; Zanette et al., 2020a,b; Weisz et al., 2020; Yang and Wang, 2019b; Modi et al., 2020; Jia et al., 2020; Ayoub et al., 2020; Zhou et al., 2021b). All these works assume that the MDP enjoys some linear representation and propose different statistical and computational complexities which depend on the dimension of the linear representation. Among these assumptions, our work falls into the category of *linear mixture MDP* which assumes that the transition dynamic can be represented as a linear combination of several basis transition probability functions (Yang and Wang, 2019b; Modi et al., 2020; Jia et al., 2020; Ayoub et al., 2020; Zhou et al., 2021b). Previous algorithms for linear mixture MDPs either suffer from a polynomial dependence on the episode horizon $H$ (Yang and Wang, 2019b; Modi et al., 2020; Jia et al., 2020; Ayoub et al., 2020; Zhou et al., 2021b; Cai et al., 2019; He et al., 2021; Zhou et al., 2021a) or do not have a computationally efficient implementation (Zhang et al., 2021c; Kim et al., 2021). Our work achieves the best of both worlds for the first time.

# 3 Preliminaries

## 3.1 Heterogeneous linear bandits

We consider the same heterogeneous linear bandits as studied in Zhou et al. (2021a). Let $\{\mathcal{D}_k\}_{k=1}^{\infty}$ be decision sets that are fixed. At each round $k$, the agent selects an action $\mathbf{a}_k \in \mathcal{D}_k$ satisfying $\|\mathbf{a}_k\|_2 \leq A$, then receives a reward $r_k$ provided by the environment. Specifically, $r_k$ is generated by $r_k = \langle \boldsymbol{\theta}^*, \mathbf{a}_k \rangle + \epsilon_k$, where $\boldsymbol{\theta}^* \in \mathbb{R}^d$ is an unknown vector, and $\epsilon_k$ is a random noise satisfying

$$\forall k, \ |\epsilon_k| \leq R, \ \mathbb{E}[\epsilon_k | \mathbf{a}_{1:k}, \epsilon_{1:k-1}] = 0, \ \mathbb{E}[\epsilon_k^2 | \mathbf{a}_{1:k}, \epsilon_{1:k-1}] \leq \sigma_k^2,$$

where $\sigma_k$ is an upper bound of the variance of the noise $\epsilon_k$ that are observable to the agent. We assume that $\sigma_k$ is $(\mathbf{a}_{1:k}, \epsilon_{1:k-1})$-measurable. The agent aims to minimize the *pseudo-regret* defined as follows:

$$\text{Regret}(K) = \sum_{k=1}^{K} [\langle \mathbf{a}_k^*, \boldsymbol{\theta}^* \rangle - \langle \mathbf{a}_k, \boldsymbol{\theta}^* \rangle], \ \text{where } \mathbf{a}_k^* = \underset{\mathbf{a} \in \mathcal{D}_k}{\operatorname{argmax}} \langle \mathbf{a}, \boldsymbol{\theta}^* \rangle.$$

## 3.2 Episodic reinforcement learning

We also study RL with linear function approximation for episodic linear mixture MDPs. We introduce the necessary definitions of MDPs here. The reader can refer to Puterman (2014) for more details.

**Episodic MDP.** We denote a homogeneous, episodic MDP by a tuple $M = M(\mathcal{S}, \mathcal{A}, H, r, \mathbb{P})$, where $\mathcal{S}$ is the state space and $\mathcal{A}$ is the action space, $H$ is the length of the episode, $r : \mathcal{S} \times \mathcal{A} \to [0, 1]$ is the deterministic reward function, and $\mathbb{P}$ is the transition probability function. For the sake of simplicity, we restrict ourselves to countable state space and finite action space. A policy $\pi = \{\pi_h\}_{h=1}^{H}$ is a collection of $H$ functions, where each of them maps a state $s$ to an action $a$.

**Value function and regret.** For $(s, a) \in \mathcal{S} \times \mathcal{A}$, we define the action-value function $Q_h^\pi(s, a)$ and (state) value function $V_h^\pi(s)$ as follows:

$$Q_h^\pi(s, a) = \mathbb{E}\left[ \sum_{h'=h}^{H} r(s_{h'}, a_{h'}) \middle| s_h = s, a_h = a, s_{h'} \sim \mathbb{P}(\cdot | s_{h'-1}, a_{h'-1}), a_{h'} = \pi_{h'}(s_{h'}) \right],$$
$$V_h^\pi(s) = Q_h^\pi(s, \pi_h(s)), \ V_{H+1}^\pi(s) = 0.$$

The optimal value function $V_h^*(\cdot)$ and the optimal action-value function $Q_h^*(\cdot, \cdot)$ are defined by $V_h^*(s) = \sup_\pi V_h^\pi(s)$ and $Q_h^*(s, a) = \sup_\pi Q_h^\pi(s, a)$, respectively. For any function $V : \mathcal{S} \to \mathbb{R}$,

we introduce the following shorthands to denote the conditional variance of $V$ at $\mathbb{P}(\cdot|s,a)$:

$$[\mathbb{P}V](s,a) = \mathbb{E}_{s'\sim\mathbb{P}(\cdot|s,a)}V(s'), \ [\mathbb{V}V](s,a) = [\mathbb{P}V^2](s,a) - ([\mathbb{P}V](s,a))^2,$$

where $V^2$ stands for the function whose value at $s$ is $V^2(s)$. Using this notation, the Bellman equations for policy $\pi$ and the Bellman optimality equation can be written as

$$Q_h^\pi(s,a) = r(s,a) + [\mathbb{P}V_{h+1}^\pi](s,a), \ Q_h^*(s,a) = r(s,a) + [\mathbb{P}V_{h+1}^*](s,a).$$

The goal is to minimize the $K$-episode regret defined as follows:

$$\text{Regret}(K) = \sum_{k=1}^K \left[V_1^*(s_1^k) - V_1^{\pi^k}(s_1^k)\right].$$

In this paper, we focus on proving high probability bounds on the regret Regret$(K)$.

In this work we make the following assumptions. The first assumption assumes that for any policy, the accumulated reward of an episode is upper bounded by 1, which has been considered in previous works (Krishnamurthy et al., 2016; Jiang and Agarwal, 2018). The accumulated reward assumption ensures that the only factor that can affect the final statistical complexity is the planning difficulty brought by the episode length, rather than the scale of the reward.

**Assumption 3.1** (Bounded total reward). For any policy $\pi$, let $\{s_h, a_h\}_{h=1}^H$ be any states and actions satisfying $a_h = \pi_h(s_h)$ and $s_{h+1} \sim \mathbb{P}(\cdot|s_h, a_h)$. Then we have $0 \le \sum_{h=1}^H r(s_h, a_h) \le 1$.

Next assumption assumes that the transition dynamic enjoys a linearized representation w.r.t. some feature mapping. We define the *linear mixture MDPs* (Jia et al., 2020; Ayoub et al., 2020; Zhou et al., 2021b) as follows.

**Assumption 3.2** (Linear mixture MDP). $M$ is an episodic $B$-bounded linear mixture MDP, such that there exists a vector $\boldsymbol{\theta}^* \in \mathbb{R}^d$ and $\boldsymbol{\phi}(\cdot|\cdot,\cdot)$ such that $\mathbb{P}(s'|s,a) = \langle \boldsymbol{\phi}(s'|s,a), \boldsymbol{\theta}^* \rangle$ for any state-action-next-state triplet $(s,a,s') \in \mathcal{S} \times \mathcal{A} \times \mathcal{S}$. Meanwhile, $\|\boldsymbol{\theta}^*\|_2 \le B$ and for any bounded function $V : \mathcal{S} \to [0,1]$ and any tuple $(s,a) \in \mathcal{S} \times \mathcal{A}$, we have

$$\|\boldsymbol{\phi}_V(s,a)\|_2 \le 1, \text{where } \boldsymbol{\phi}_V(s,a) = \sum_{s' \in \mathcal{S}} \boldsymbol{\phi}(s'|s,a)V(s').$$

Lastly, for any $V : \mathcal{S} \to [0,1]$, $\boldsymbol{\phi}_V$ can be calculated efficiently within $\mathcal{O}$ time.

**Remark 3.3.** A key property of linear mixture MDP is that for any function $V : \mathcal{S} \to \mathbb{R}$ and any state-action pair $(s,a)$, the conditional expectation of $V$ over $\mathbb{P}(\cdot|s,a)$ is a linear function of $\boldsymbol{\theta}^*$, i.e., $[\mathbb{P}V](s,a) = \langle \boldsymbol{\phi}_V(s,a), \boldsymbol{\theta}^* \rangle$. Meanwhile, the conditional variance of $V$ over $\mathbb{P}(\cdot|s,a)$ is a quadratic function of $\boldsymbol{\theta}^*$, i.e., $[\mathbb{V}V](s,a) = \langle \boldsymbol{\phi}_{V^2}(s,a), \boldsymbol{\theta}^* \rangle - [\langle \boldsymbol{\phi}_V(s,a), \boldsymbol{\theta}^* \rangle]^2$.

**Remark 3.4.** For a general class of $\boldsymbol{\phi}$, $\boldsymbol{\phi}_V(s,a) : \mathcal{S} \times \mathcal{A} \to \mathbb{R}^d$ can be computed efficiently for any $(s,a) \in \mathcal{S} \times \mathcal{A}$ if $V : \mathcal{S} \to \mathbb{R}$ can be computed efficiently. For instance, $\boldsymbol{\phi}(s'|s,a) = \mathbf{e}_{s',s,a} \in \mathbb{R}^{|\mathcal{S}|^2|\mathcal{A}|}$ and $\boldsymbol{\phi}(s'|s,a) = \boldsymbol{\psi}(s') \odot \boldsymbol{\xi}(s,a)$, where $\boldsymbol{\psi}, \boldsymbol{\xi}$ are two sub feature functions. More discussions are referred to Zhou et al. (2021a).

## 4 Computationally efficient variance-aware linear bandits

In this section, we propose our algorithm WeightedOFUL$^+$ in Algorithm 1 for the heterogeneous linear bandits introduced in Section 3.1. WeightedOFUL$^+$ adopts the *weighted ridge regression* estimator used by WeightedOFUL (Zhou et al., 2021a), but uses a refined weight. It first computes a weighted estimate of $\boldsymbol{\theta}^*$, denoted by $\widehat{\boldsymbol{\theta}}_k$, based on previous contexts and rewards, where the weights $\bar{\sigma}_k$ are computed by the noise variance $\sigma_k$. Then WeightedOFUL$^+$ constructs the confidence set of $\boldsymbol{\theta}^*$, denoted by $\widehat{\mathcal{C}}_k$, estimates the reward $\langle \mathbf{a}, \boldsymbol{\theta} \rangle$ for $\boldsymbol{\theta} \in \widehat{\mathcal{C}}_k$, and selects the arm that maximizes the estimated reward optimistically. The selection rule of $\mathbf{a}_k$ is identical to selecting the best arm w.r.t. to their upper confidence bound, i.e., $\mathbf{a}_k \leftarrow \text{argmax}_{\mathbf{a} \in \mathcal{D}_k} \langle \mathbf{a}, \widehat{\boldsymbol{\theta}}_k \rangle + \widehat{\beta}_k \|\mathbf{a}\|_{\widehat{\boldsymbol{\Sigma}}_k^{-1}}$ (Li et al., 2010). WeightedOFUL$^+$ constructs $\bar{\sigma}_k^2$ as the maximization of the variance, a constant, and the uncertainty $\|\mathbf{a}_k\|_{\widehat{\boldsymbol{\Sigma}}_k^{-1}}$, as defined in (4.1). Note that Zhou et al. (2021a) proposed a *variance-aware* weight $\bar{\sigma}_k = \max\{\sigma_k, \alpha\}$ for heterogeneous linear bandits. The additional uncertainty term in our weight

---
**Algorithm 1** WeightedOFUL$^+$

---
**Require:** Regularization parameter $\lambda > 0$, and $B$, an upper bound on the $\ell_2$-norm of $\boldsymbol{\theta}^*$, confidence
    radius $\widehat{\beta}_k$, variance parameters $\alpha, \gamma$
1: $\widehat{\boldsymbol{\Sigma}}_1 \leftarrow \lambda\mathbf{I}, \widehat{\mathbf{b}}_1 \leftarrow \mathbf{0}, \widehat{\boldsymbol{\theta}}_1 \leftarrow \mathbf{0}, \widehat{\beta}_1 = \sqrt{\lambda}B$
2: **for** $k = 1, \ldots, K$ **do**
3:     Let $\widehat{\mathcal{C}}_k \leftarrow \{\boldsymbol{\theta} : \|\widehat{\boldsymbol{\Sigma}}_k^{1/2}(\boldsymbol{\theta} - \widehat{\boldsymbol{\theta}}_k)\|_2 \leq \widehat{\beta}_k\}$, observe $\mathcal{D}_k$
4:     Set $(\mathbf{a}_k, \boldsymbol{\theta}_k) \leftarrow \operatorname{argmax}_{\mathbf{a}\in\mathcal{D}_k, \boldsymbol{\theta}\in\widehat{\mathcal{C}}_k} \langle\mathbf{a}, \boldsymbol{\theta}\rangle$
5:     Observe $(r_k, \sigma_k)$, set $\bar{\sigma}_k$ as

$$\bar{\sigma}_k \leftarrow \max\{\sigma_k, \alpha, \gamma\|\mathbf{a}_k\|_{\widehat{\boldsymbol{\Sigma}}_k^{-1}}^{1/2}\} \tag{4.1}$$

6:     $\widehat{\boldsymbol{\Sigma}}_{k+1} \leftarrow \widehat{\boldsymbol{\Sigma}}_k + \mathbf{a}_k\mathbf{a}_k^\top/\bar{\sigma}_k^2, \widehat{\mathbf{b}}_{k+1} \leftarrow \widehat{\mathbf{b}}_k + r_k\mathbf{a}_k/\bar{\sigma}_k^2, \widehat{\boldsymbol{\theta}}_{k+1} \leftarrow \widehat{\boldsymbol{\Sigma}}_{k+1}^{-1}\widehat{\mathbf{b}}_{k+1}$
7: **end for**

---

enables us to build a tighter confidence set $\widehat{\mathcal{C}}_k$ since the uncertainty of arms can be leveraged by a tighter Bernstein-type concentration inequality (See Section 4.1 for more details). Meanwhile, we notice that He et al. (2022) proposed a pure *uncertainty-aware* weight $\bar{\sigma}_k = \max\{\alpha, \gamma\|\mathbf{a}_k\|_{\widehat{\boldsymbol{\Sigma}}_k^{-1}}^{1/2}\}$ to deal with the corruption in contextual linear bandits, which serves a different purpose compared to our setting (there is no corruption in our bandit model).

The following theorem gives the regret bound of WeightedOFUL$^+$.

**Theorem 4.1.** Let $0 < \delta < 1$. Suppose that for all $k \geq 1$ and all $\mathbf{a} \in \mathcal{D}_k, \langle\mathbf{a}, \boldsymbol{\theta}^*\rangle \in [-1, 1]$, $\|\boldsymbol{\theta}^*\|_2 \leq B$ and $\{\widehat{\beta}_k\}_{k\geq 1}$ are set to

$$\widehat{\beta}_k = 12\sqrt{d\log(1 + kA^2/(\alpha^2 d\lambda))\log(32(\log(\gamma^2/\alpha) + 1)k^2/\delta)}$$
$$+ 30\log(32(\log(\gamma^2/\alpha) + 1)k^2/\delta)R/\gamma^2 + \sqrt{\lambda}B. \tag{4.2}$$

Then with probability at least $1 - \delta$, the regret of WeightedOFUL$^+$ is bounded by

$$\text{Regret}(K) \leq 4d\iota + 4d\gamma^2\widehat{\beta}_K\iota + 4\widehat{\beta}_K\sqrt{\sum_{k=1}^K \sigma_k^2 + K\alpha^2}\sqrt{d\iota}, \tag{4.3}$$

where $\iota = \log(1 + KA^2/(d\lambda\alpha^2))$. Moreover, setting $\alpha = 1/\sqrt{K}, \gamma = R^{1/2}/d^{1/4}$ and $\lambda = d/B^2$ yields a high probability regret $\text{Regret}(K) = \widetilde{O}(d\sqrt{\sum_{k=1}^K \sigma_k^2} + dR + d)$.

**Remark 4.2.** Treating $R$ as a constant, the regret of WeightedOFUL$^+$ becomes $\widetilde{O}(d\sqrt{\sum_{k=1}^K \sigma_k^2} + d)$. It strictly outperforms the $\widetilde{O}(d\sqrt{\sum_{k=1}^K \sigma_k^2} + \sqrt{dK} + d)$ regret achieved in Zhou et al. (2021a). Compared with the $\widetilde{O}(d^{4.5}\sqrt{\sum_{k=1}^K \sigma_k^2} + d^5)$ regret by VOFUL (Zhang et al., 2021c) and the $\widetilde{O}(d^{1.5}\sqrt{\sum_{k=1}^K \sigma_k^2} + d^2)$ regret by VOFUL2 (Kim et al., 2021), the regret of WeightedOFUL$^+$ has a better dependence on $d$, and WeightedOFUL$^+$ is computationally efficient. It is worth noting that both VOFUL and VOFUL2 do not need to know the variance $\sigma_k^2$ while our algorithm does. Whether there exists an algorithm that can achieve the $\widetilde{O}(d\sqrt{\sum_{k=1}^K \sigma_k^2} + \sqrt{dK} + d)$ regret without knowing the variance information remains an open problem.

## 4.1 Proof sketch

We give a proof sketch of Theorem 4.1 along with two key lemmas that are pivotal to obtain the improved regret. To begin with, we first show that $\boldsymbol{\theta}^*$ belongs to confidence balls centering at $\widehat{\boldsymbol{\theta}}_k$ with radius $\widehat{\beta}_k$. This can be proved by the following lemma, which is an improved version of the Bernstein-type self-normalized martingale inequality proposed by Zhou et al. (2021a).

**Lemma 4.3.** Let $\{\mathcal{G}_k\}_{k=1}^\infty$ be a filtration, and $\{\mathbf{x}_k, \eta_k\}_{k\geq 1}$ be a stochastic process such that $\mathbf{x}_k \in \mathbb{R}^d$ is $\mathcal{G}_k$-measurable and $\eta_k \in \mathbb{R}$ is $\mathcal{G}_{k+1}$-measurable. Let $L, \sigma, \lambda, \epsilon > 0, \boldsymbol{\mu}^* \in \mathbb{R}^d$. For $k \geq 1$, let

$y_k = \langle \boldsymbol{\mu}^*, \mathbf{x}_k \rangle + \eta_k$ and suppose that $\eta_k, \mathbf{x}_k$ also satisfy

$$\mathbb{E}[\eta_k|\mathcal{G}_k] = 0, \ \mathbb{E}[\eta_k^2|\mathcal{G}_k] \leq \sigma^2, \ |\eta_k| \leq R, \ \|\mathbf{x}_k\|_2 \leq L. \tag{4.4}$$

For $k \geq 1$, let $\mathbf{Z}_k = \lambda \mathbf{I} + \sum_{i=1}^{k} \mathbf{x}_i \mathbf{x}_i^\top$, $\mathbf{b}_k = \sum_{i=1}^{k} y_i \mathbf{x}_i$, $\boldsymbol{\mu}_k = \mathbf{Z}_k^{-1} \mathbf{b}_k$, and

$$\beta_k = 12\sqrt{\sigma^2 d \log(1 + kL^2/(d\lambda)) \log(32(\log(R/\epsilon)+1)k^2/\delta)}$$
$$+ 24\log(32(\log(R/\epsilon)+1)k^2/\delta) \max_{1 \leq i \leq k}\{|\eta_i|\min\{1, \|\mathbf{x}_i\|_{\mathbf{Z}_{i-1}^{-1}}\}\} + 6\log(32(\log(R/\epsilon)+1)k^2/\delta)\epsilon.$$

Then, for any $0 < \delta < 1$, we have with probability at least $1 - \delta$ that,

$$\forall k \geq 1, \ \left\|\sum_{i=1}^{k}\mathbf{x}_i\eta_i\right\|_{\mathbf{Z}_k^{-1}} \leq \beta_k, \ \|\boldsymbol{\mu}_k - \boldsymbol{\mu}^*\|_{\mathbf{Z}_k} \leq \beta_k + \sqrt{\lambda}\|\boldsymbol{\mu}^*\|_2,$$

Note that $\widehat{\boldsymbol{\theta}}_k$ can be regarded as $\boldsymbol{\mu}_{k-1}$ in Lemma 4.3 with $\epsilon = R/\gamma^2$, $\mathbf{x}_k = \mathbf{a}_k/\bar{\sigma}_k$, $y_k = r_k/\bar{\sigma}_k$, $\eta_k = \epsilon_k/\bar{\sigma}_k$, $\mathbf{Z}_k = \widehat{\boldsymbol{\Sigma}}_{k+1}$ and $\boldsymbol{\mu}^* = \boldsymbol{\theta}^*$. With the help of the weight $\bar{\sigma}_k$, the variance of $\eta_k$ is upper bounded by 1 (since $\bar{\sigma}_k \geq \sigma_k$) and $|\eta_k|\min\{1, \|\mathbf{x}_k\|_{\mathbf{Z}_k^{-1}}\} \leq |\epsilon_k|\|\mathbf{a}_k\|_{\widehat{\boldsymbol{\Sigma}}_k^{-1}}/\bar{\sigma}_k^2 \leq R/\gamma^2$ (since $\bar{\sigma}_k^2 \geq \gamma^2\|\mathbf{a}_k\|_{\widehat{\boldsymbol{\Sigma}}_k^{-1}}$). Therefore, by Lemma 4.3, w.h.p. $\boldsymbol{\theta}^* \in \widehat{\mathcal{C}}_k$. Following the standard procedure to bound the regret of the optimistic algorithm (Abbasi-Yadkori et al., 2011), we have

$$\text{Regret}(K) = \sum_{k=1}^{K}\langle \mathbf{a}_k^* - \mathbf{a}_k, \boldsymbol{\theta}^*\rangle \leq 2\sum_{k=1}^{K}\min\{1, \widehat{\beta}_k\|\mathbf{a}_k\|_{\widehat{\boldsymbol{\Sigma}}_k^{-1}}\}. \tag{4.5}$$

Next lemma gives an upper bound of (4.5).

**Lemma 4.4.** Let $\{\sigma_k, \widehat{\beta}_k\}_{k\geq 1}$ be a sequence of non-negative numbers, $\alpha, \gamma > 0$, $\{\mathbf{a}_k\}_{k\geq 1} \subset \mathbb{R}^d$ and $\|\mathbf{a}_k\|_2 \leq A$. Let $\{\bar{\sigma}_k\}_{k\geq 1}$ and $\{\widehat{\boldsymbol{\Sigma}}_k\}_{k\geq 1}$ be (recursively) defined as follows: $\widehat{\boldsymbol{\Sigma}}_1 = \lambda \mathbf{I}$,

$$\forall k \geq 1, \ \bar{\sigma}_k = \max\{\sigma_k, \alpha, \gamma\|\mathbf{a}_k\|_{\widehat{\boldsymbol{\Sigma}}_k^{-1}}^{1/2}\}, \ \widehat{\boldsymbol{\Sigma}}_{k+1} = \widehat{\boldsymbol{\Sigma}}_k + \mathbf{a}_k\mathbf{a}_k^\top/\bar{\sigma}_k^2.$$

Let $\iota = \log(1 + KA^2/(d\lambda\alpha^2))$. Then we have

$$\sum_{k=1}^{K}\min\left\{1, \widehat{\beta}_k\|\mathbf{a}_k\|_{\widehat{\boldsymbol{\Sigma}}_k^{-1}}\right\} \leq 2d\iota + 2\max_{k\in[K]}\widehat{\beta}_k\gamma^2 d\iota + 2\sqrt{d\iota}\sqrt{\sum_{k=1}^{K}\widehat{\beta}_k^2(\sigma_k^2 + \alpha^2)}.$$

By Lemma 4.4, the regret of WeightedOFUL$^+$ can be bounded by

$$\text{Regret}(K) = \widetilde{O}\big(d + \sqrt{d}\widehat{\beta}_K\big(\sqrt{\textstyle\sum_{k=1}^{K}\sigma_k^2 + K\alpha^2} + \sqrt{d}\gamma^2\big)\big) \tag{4.6}$$

with $\widehat{\beta}_K = \widetilde{O}(\sqrt{d} + R/\gamma^2 + \sqrt{\lambda}B)$, which finishes the proof.

Here we compare the regret of WeightedOFUL$^+$ and the regret of WeightedOFUL (Zhou et al., 2021a) (which chooses the weight $\bar{\sigma}_k = \max\{\sigma_k, \alpha\}$) to see where the improvement comes from. In particular, Zhou et al. (2021a) proved the regret of WeightedOFUL as follows

$$\text{Regret}(K) = \widetilde{O}\big(d + \sqrt{d}\widehat{\beta}_K^\alpha\sqrt{\textstyle\sum_{k=1}^{K}\sigma_k^2 + K\alpha^2}\big) \tag{4.7}$$

with $\widehat{\beta}_K^\alpha = \widetilde{O}(\sqrt{d} + R/\alpha + \sqrt{\lambda}B)$. At the first glance, both (4.6) and (4.7) have a $\sqrt{K\alpha^2}$ term. However, thanks to the design of $\bar{\sigma}_k$ and Lemma 4.3, the $\sqrt{K\alpha^2}$ term in (4.6) can be shaved by choosing a small enough $\alpha$, while this term in (4.7) cannot be shaved due to the existence of a $R/\alpha$ term in $\widehat{\beta}_K^\alpha$.

# 5 Computationally efficient horizon-free RL for linear mixture MDPs

In this section, we propose a horizon-free RL algorithm HF-UCRL-VTR$^+$ in Algorithm 2. We leave the discussion of the computational complexity of HF-UCRL-VTR$^+$ to Appendix B.

HF-UCRL-VTR$^+$ follows the *value targeted regression (VTR)* framework proposed by Jia et al. (2020); Ayoub et al. (2020) to learn the linear mixture MDP. In detail, following the observation in

---

**Algorithm 2** HF-UCRL-VTR$^+$

---

**Require:** Regularization parameter $\lambda$, an upper bound $B$ of the $\ell_2$-norm of $\boldsymbol{\theta}^*$, confidence radius $\{\widehat{\beta}_k\}_{k\geq 1}$, level $M$, variance parameters $\alpha, \gamma$, $\overline{[M]} = \{0, \ldots, M-1\}$

1: For $m \in \overline{[M]}$, set $\widehat{\boldsymbol{\theta}}_{1,m} \leftarrow \mathbf{0}$, $\widetilde{\boldsymbol{\Sigma}}_{0,H+1,m} \leftarrow \lambda\mathbf{I}$, $\widetilde{\mathbf{b}}_{0,H+1,m} \leftarrow \mathbf{0}$. Set $V_{1,H+1}(\cdot) \leftarrow 0$
2: **for** $k = 1, \ldots, K$ **do**
3:     **for** $h = H, \ldots, 1$ **do**
4:         Set $Q_{k,h}(\cdot, \cdot) \leftarrow \left[ r(\cdot, \cdot) + \langle \widehat{\boldsymbol{\theta}}_{k,0}, \boldsymbol{\phi}_{V_{k,h+1}}(\cdot, \cdot) \rangle + \widehat{\beta}_k \left\| \widehat{\boldsymbol{\Sigma}}_{k,0}^{-1/2} \boldsymbol{\phi}_{V_{k,h+1}}(\cdot, \cdot) \right\|_2 \right]_{[0,1]}$
5:         Set $\pi_h^k(\cdot) \leftarrow \operatorname{argmax}_{a \in \mathcal{A}} Q_{k,h}(\cdot, a)$
6:         Set $V_{k,h}(\cdot) \leftarrow \max_{a \in \mathcal{A}} Q_{k,h}(\cdot, a)$
7:     **end for**
8:     Receive $s_1^k$. For $m \in \overline{[M]}$, set $\widetilde{\boldsymbol{\Sigma}}_{k,1,m} \leftarrow \widetilde{\boldsymbol{\Sigma}}_{k-1,H+1,m}$
9:     **for** $h = 1, \ldots, H$ **do**
10:         Take action $a_h^k \leftarrow \pi_h^k(s_h^k)$, receive $s_{h+1}^k \sim \mathbb{P}(\cdot | s_h^k, a_h^k)$.
11:         For $m \in \overline{[M]}$, denote $\boldsymbol{\phi}_{k,h,m} = \boldsymbol{\phi}_{V_{k,h+1}^{2^m}}(s_h^k, a_h^k)$.
12:         Set $\{\bar{\sigma}_{k,h,m}\}_{m \in \overline{[M]}} \leftarrow$ Algorithm 3($\{\boldsymbol{\phi}_{k,h,m}, \widehat{\boldsymbol{\theta}}_{k,m}, \widetilde{\boldsymbol{\Sigma}}_{k,h,m}, \widehat{\boldsymbol{\Sigma}}_{k,m}\}_{m \in \overline{[M]}}, \widehat{\beta}_k, \alpha, \gamma$)
13:         For $m \in \overline{[M]}$, set $\widetilde{\boldsymbol{\Sigma}}_{k,h+1,m} \leftarrow \widetilde{\boldsymbol{\Sigma}}_{k,h,m} + \boldsymbol{\phi}_{k,h,m}\boldsymbol{\phi}_{k,h,m}^\top / \bar{\sigma}_{k,h,m}^2$
14:         For $m \in \overline{[M]}$, set $\widetilde{\mathbf{b}}_{k,h+1,m} \leftarrow \widetilde{\mathbf{b}}_{k,h,m} + \boldsymbol{\phi}_{k,h,m}V_{k,h+1}^{2^m}(s_{h+1}^k) / \bar{\sigma}_{k,h,m}^2$
15:     **end for**
16:     For $m \in \overline{[M]}$, set $\widehat{\boldsymbol{\Sigma}}_{k+1,m} \leftarrow \widetilde{\boldsymbol{\Sigma}}_{k,H+1,m}, \widehat{\mathbf{b}}_{k+1,m} \leftarrow \widetilde{\mathbf{b}}_{k,H+1,m}, \widehat{\boldsymbol{\theta}}_{k+1,m} \leftarrow \widehat{\boldsymbol{\Sigma}}_{k+1,m}^{-1} \widehat{\mathbf{b}}_{k+1,m}$
17: **end for**

---

Remark 3.3, VTR estimates $\boldsymbol{\theta}^*$ by solving a regression problem over predictors/contexts $\boldsymbol{\phi}_{k,h,0} = \boldsymbol{\phi}_{V_{k,h+1}}(s_h^k, a_h^k)$ and responses $V_{k,h+1}(s_{h+1}^k)$. Specifically, HF-UCRL-VTR$^+$ takes the estimate $\widehat{\boldsymbol{\theta}}_{k,0}$ as the solution to the following weighted regression problem:

$$\widehat{\boldsymbol{\theta}}_{k,0} = \underset{\boldsymbol{\theta} \in \mathbb{R}^d}{\operatorname{argmin}} \, \lambda \|\boldsymbol{\theta}\|_2^2 + \sum_{j=1}^{k-1} \sum_{h=1}^{H} \left[ \langle \boldsymbol{\phi}_{j,h,0}, \boldsymbol{\theta} \rangle - V_{j,h+1}(s_{h+1}^j) \right]^2 / \bar{\sigma}_{j,h,0}^2, \quad (5.1)$$

where $\bar{\sigma}_{j,h,0}$ is the upper bound of the conditional variance $[\mathbb{V}V_{k,h+1}](s_h^k, a_h^k)$. Such a weighted regression scheme has been adapted by UCRL-VTR$^+$ in Zhou et al. (2021a). With $\widehat{\boldsymbol{\theta}}_{k,0}$, HF-UCRL-VTR$^+$ then constructs the optimistic estimates $Q_{k,h}$ (resp. $V_{k,h}$) of the optimal value functions $Q_h^*$ (resp. $V_h^*$) and takes actions optimistically. Note that $\widehat{\boldsymbol{\theta}}_{k,0}$ is updated at the end of each episode. We highlight several improved algorithm designs of HF-UCRL-VTR$^+$ compared to UCRL-VTR$^+$ as follows.

**Improved weighted linear regression estimator.** HF-UCRL-VTR$^+$ sets $\bar{\sigma}_{j,h,0}$ similar to the weight used in WeightedOFUL$^+$. Assuming that the conditional variance $[\mathbb{V}V_{k,h+1}](s_h^k, a_h^k)$ can be computed for any value function $V$ and state action pair $(s, a)$, then the weight can be set as

$$\bar{\sigma}_{k,h,0}^2 = \max\{[\mathbb{V}V_{k,h+1}](s_h^k, a_h^k), \alpha^2, \gamma^2 \|\widetilde{\boldsymbol{\Sigma}}_{k,h,0}^{-1/2} \boldsymbol{\phi}_{k,h,0}\|_2\},$$

where $\widetilde{\boldsymbol{\Sigma}}_{k,h,0}$ is the weighted sample covariance matrix of $\boldsymbol{\phi}_{k,h,0}$ up to $k$-th episode and $h$-th stage. However, the true variance is not accessible since $\mathbb{P}(\cdot | s, a)$ is unknown. Therefore, HF-UCRL-VTR$^+$ replaces $[\mathbb{V}V_{k,h+1}](s_h^k, a_h^k)$ with its estimate $[\bar{\mathbb{V}}_{k,0}V_{k,h+1}](s_h^k, a_h^k)$ and an error bound $E_{k,h,0}$ satisfying $[\bar{\mathbb{V}}_{k,0}V_{k,h+1}](s_h^k, a_h^k) + E_{k,h,0} \geq [\mathbb{V}V_{k,h+1}](s_h^k, a_h^k)$ with high probability. Thanks to the fact that $[\mathbb{V}V_{k,h+1}](s_h^k, a_h^k)$ is a quadratic function of $\boldsymbol{\theta}^*$ as illustrated in Remark 3.3, $[\bar{\mathbb{V}}_{k,0}V_{k,h+1}](s_h^k, a_h^k)$ can be estimated as follows:

$$[\bar{\mathbb{V}}_{k,0}V_{k,h+1}](s_h^k, a_h^k) = \left[ \langle \boldsymbol{\phi}_{k,h,1}, \widehat{\boldsymbol{\theta}}_{k,1} \rangle \right]_{[0,1]} - \left[ \langle \boldsymbol{\phi}_{k,h,0}, \widehat{\boldsymbol{\theta}}_{k,0} \rangle \right]_{[0,1]}^2, \quad (5.2)$$

where $\widehat{\boldsymbol{\theta}}_{k,1}$ is the solution to some regression problem over predictors/contexts $\boldsymbol{\phi}_{k,h,1} = \boldsymbol{\phi}_{V_{k,h+1}^2}(s_h^k, a_h^k)$ and responses $V_{k,h+1}^2(s_{h+1}^k)$.

**Higher-order moment regression.** To obtain a better estimate, it is natural to set $\widehat{\boldsymbol{\theta}}_{k,1}$ as the solution to the weighted regression problem on $\boldsymbol{\phi}_{k,h,1}$ and $V_{k,h+1}^2(s_{h+1}^k)$ with weight $\bar{\sigma}_{k,h,1}$. Here $\bar{\sigma}_{k,h,1}$ is

---

**Algorithm 3** High-order moment estimator (HOME)

---

**Require:** Features $\{\phi_{k,h,m}\}_{m\in\overline{[M]}}$, vector estimators $\{\widehat{\theta}_{k,m}\}_{m\in\overline{[M]}}$, covariance matrix $\{\widetilde{\Sigma}_{k,h,m},\widehat{\Sigma}_{k,m}\}_{m\in\overline{[M]}}$, confidence radius $\widehat{\beta}_k,\alpha,\gamma$
1: **for** $m=0,\dots,M-2$ **do**
2:   Set $[\bar{\mathbb{V}}_{k,m}V_{k,h+1}^{2^m}](s_h^k,a_h^k)\leftarrow [\langle\phi_{k,h,m+1},\widehat{\theta}_{k,m+1}\rangle]_{[0,1]}-[\langle\phi_{k,h,m},\widehat{\theta}_{k,m}\rangle]_{[0,1]}^2$
3:   Set $E_{k,h,m}\leftarrow\min\{1,2\widehat{\beta}_k\|\widehat{\Sigma}_{k,m}^{-1/2}\phi_{k,h,m}\|_2\}+\min\{1,\widehat{\beta}_k\|\widehat{\Sigma}_{k,m+1}^{-1/2}\phi_{k,h,m+1}\|_2\}$
4:   Set $\bar{\sigma}_{k,h,m}^2\leftarrow\max\{[\bar{\mathbb{V}}_{k,m}V_{k,h+1}^{2^m}](s_h^k,a_h^k)+E_{k,h,m},\alpha^2,\gamma^2\|\widetilde{\Sigma}_{k,h,m}^{-1/2}\phi_{k,h,m}\|_2\}$
5: **end for**
6: Set $\bar{\sigma}_{k,h,M-1}^2\leftarrow\max\{1,\alpha^2,\gamma^2\|\widetilde{\Sigma}_{k,h,M-1}^{-1/2}\phi_{k,h,M-1}\|_2\}$
**Ensure:** $\{\bar{\sigma}_{k,h,m}\}_{m\in\overline{[M]}}$

---

constructed in a similar way to $\bar{\sigma}_{k,h,0}$, which relies on the conditional variance of $[\mathbb{V}V_{k,h+1}^2](s_h^k,a_h^k)$. By repeating this process, we recursively estimate the conditional $2^m$-th moment of $V_{k,h+1}$ by its variance, which is the conditional $2^{m+1}$-th moment of $V_{k,h+1}$. It is worth noting that the idea of high-order recursive estimation has been used in Li et al. (2020) and later in Zhang et al. (2021b,c) to achieve horizon-free regret/sample complexity guarantees. Similar recursive analysis also appeared in Lattimore and Hutter (2012).

The estimated conditional moment $[\bar{\mathbb{V}}_{k,m}V_{k,h+1}^{2^m}](s_h^k,a_h^k)$ relies on $\widehat{\theta}_{k,m+1}$ and $\widehat{\theta}_{k,m}$, and $\langle\phi_{k,h,m+1},\widehat{\theta}_{k,m+1}\rangle$ serves as the estimate of the higher-moment $[\mathbb{V}V_{k,h+1}^{2^{m+1}}](s_h^k,a_h^k)$. The detailed constructions for the high-order moment estimator are summarized in Algorithm 3.

We provide the regret bound for HF-UCRL-VTR$^+$ here.

**Theorem 5.1.** Set $M=\log(3KH)/\log 2$. For any $\delta>0$, set $\{\widehat{\beta}_k\}_{k\geq 1}$ as

$$\widehat{\beta}_k=12\sqrt{d\log(1+kH/(\alpha^2d\lambda))\log(32(\log(\gamma^2/\alpha)+1)k^2H^2/\delta)}$$
$$+30\log(32(\log(\gamma^2/\alpha)+1)k^2H^2/\delta)/\gamma^2+\sqrt{\lambda}B, \tag{5.3}$$

then with probability at least $1-(2M+1)\delta$, the regret of Algorithm 2 is bounded by

$$\text{Regret}(K)\leq 12(8d\iota+8\widehat{\beta}_K\gamma^2d\iota+4\widehat{\beta}_K\sqrt{d\iota}\sqrt{Md\iota/2+KH\alpha^2}+\sqrt{Md\iota\zeta}+\zeta)$$
$$+864\max\{8\widehat{\beta}_K^2d\iota,\zeta\}+Md\iota/2+[\sqrt{2\log(1/\delta)}+32\max\{2\widehat{\beta}_K\sqrt{2d\iota},\sqrt{\zeta}\}]\sqrt{K},$$

where $\iota=\log(1+KH/(d\lambda\alpha^2))$, $\zeta=4\log(4\log(KH)/\delta)$. Moreover, setting $\alpha=\sqrt{d/(KH)},\gamma=1/d^{1/4}$ and $\lambda=d/B^2$ yields a high-probability regret $\text{Regret}(K)=\widetilde{O}(d\sqrt{K}+d^2)$.

**Remark 5.2.** The regret of HF-UCRL-VTR$^+$ is strictly better than that of VOFUL $\widetilde{O}(d^{4.5}\sqrt{K}+d^9)$ Zhang et al. (2021c), and it matches the regret of VOFUL2 (Kim et al., 2021). More importantly, HF-UCRL-VTR$^+$ is computationally efficient, while there is no efficient implementation of VO-FUL/VOFUL2.

Next theorem provides the regret lower bound and suggests that the regret obtained by HF-UCRL-VTR$^+$ is near-optimal. The lower bound is proved by constructing hard-instances of linear mixture MDPs following Zhou et al. (2021b,a); Zhang et al. (2021a).

**Theorem 5.3.** Let $B>1$. Then for any algorithm, when $K\geq\max\{3d^2,(d-1)/(192(B-1))\}$, there exists a $B$-bounded linear mixture MDP satisfying Assumptions 3.1 and 3.2 such that its expected regret $\mathbb{E}[\text{Regret}(K)]$ is lower bounded by $d\sqrt{K}/(16\sqrt{3})$.

**Remark 5.4.** When specialized to tabular MDPs where $d=|\mathcal{S}|^2|\mathcal{A}|$, HF-UCRL-VTR$^+$ yields a horizon-free regret $\widetilde{O}(|\mathcal{S}|^2|\mathcal{A}|\sqrt{K}+|\mathcal{S}|^4|\mathcal{A}|^2)$. Although the regret does not match the near-optimal result $\widetilde{O}(\sqrt{|\mathcal{S}||\mathcal{A}|K}+|\mathcal{S}|^2|\mathcal{A}|)$ (Zhang et al., 2021b), it is not surprising since HF-UCRL-VTR$^+$ is designed for a more general MDP class. We leave the design of algorithms that achieve near-optimal regret for both linear mixture MDPs and tabular MDPs simultaneously as a future work.

We conduct some numerical experiments to suggest the validity of HF-UCRL-VTR$^+$ in Appendix A.

## 6 Conclusion

In this work, we propose a new weighted linear regression estimator that adapts *variance-uncertainty-aware* weights, which can be applied to both heterogeneous linear bandits and linear mixture MDPs. For heterogeneous linear bandits, our WeightedOFUL$^+$ algorithm achieves an $\widetilde{O}(d\sqrt{\sum_{k=1}^{K}\sigma_k^2}+d)$ regret in the first $K$ rounds. For linear mixture MDPs, our HF-UCRL-VTR$^+$ algorithm achieves the near-optimal $\widetilde{O}(d\sqrt{K}+d^2)$ regret. Both of our algorithms are computationally efficient and yield the state-of-the-arts regret results.

## Acknowledgments and Disclosure of Funding

We thank the anonymous reviewers for their helpful comments. DZ and QG are partially supported by the National Science Foundation CAREER Award 1906169 and the Sloan Research Fellowship. The views and conclusions contained in this paper are those of the authors and should not be interpreted as representing any funding agencies.

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
