# A  Experiments

In this section, we conduct some numerical experiments to validate our theory. The code and data for our experiments can be found on Github [3].

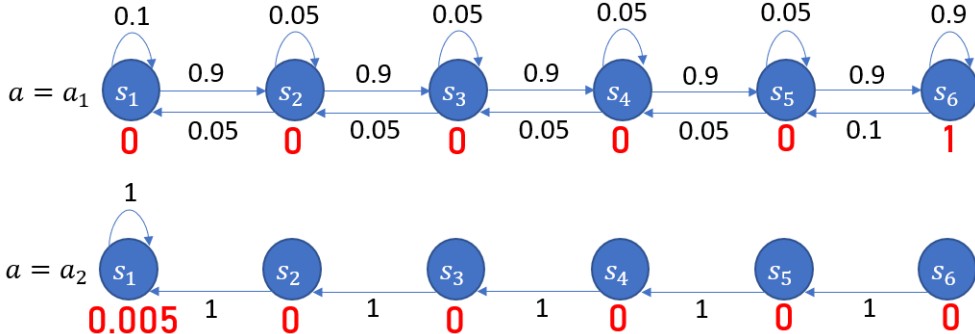

Figure 1: Transition dynamic and reward of RiverSwim. The black numbers denote the transition probability, the red numbers denote the reward. The first row represents the transition probability with action $a_1$, and the second row represents the transition probability with action $a_2$.

**Environment** We follow the RiverSwim environment introduced by Strehl and Littman (2008) and considered by Ayoub et al. (2020). The state space is $\mathcal{S} = \{s_1, \ldots, s_6\}$, the action space is $\mathcal{A} = \{a_1, a_2\}$. The transition dynamic and reward are demonstrated by Figure 1. The length of the episode $H = 100, 500$. The initial state is $s_1$. The RiverSwim MDP is a linear mixture MDP with $d = 3$ and a feature mapping $\phi(s'|s, a) \in \mathbb{R}^3$. Here

$$\phi(s'|s_1, a_1) = \mathbb{1}(s' = s_1)(0, 1, 1) + \mathbb{1}(s' = s_2)(1, 0, 0)$$
$$\forall 2 \le i \le 5, \phi(s'|s_i, a_1) = \mathbb{1}(s' = s_{i-1})(0, 0, 1) + \mathbb{1}(s' = s_i)(0, 1, 0) + \mathbb{1}(s' = s_{i+1})(1, 0, 0)$$
$$\phi(s'|s_6, a_1) = \mathbb{1}(s' = s_5)(0, 1, 1) + \mathbb{1}(s' = s_6)(1, 0, 0)$$
$$\phi(s'|s_1, a_2) = \mathbb{1}(s' = s_1)(1, 1, 1)$$
$$\forall 2 \le i \le 6, \phi(s'|s_i, a_2) = \mathbb{1}(s' = s_{i-1})(1, 1, 1)$$

The transition dynamic $\mathbb{P} = \langle \phi/3, \boldsymbol{\theta} \rangle$ with $\boldsymbol{\theta} = (2.7, 0.15, 0.15)$.

**Benchmarks** We compare the following algorithms.

- *HF-UCRL-VTR$^+$ with $M = 4$ (level parameter of HOME)*
- *HF-UCRL-VTR$^+$ with $M = 2$ (level parameter of HOME)*
- *UCRL-VTR$^+$* (Zhou et al., 2021a)
- *UCRL-VTR* (Jia et al., 2020; Ayoub et al., 2020)
- *Q-learning* (Jin et al., 2018)
- *Random*: Randomly select actions

The hyperparameters of each algorithm are grid searched for the best performance. For HF-UCRL-VTR$^+$ with $M = 4$, HF-UCRL-VTR$^+$ with $M = 2$, UCRL-VTR$^+$ and UCRL-VTR, we set the regularization parameter $\lambda = 0.001$, and the confidence radius parameter $\beta_k = 1$. For HF-UCRL-VTR$^+$ with $M = 4$, HF-UCRL-VTR$^+$ with $M = 2$, we set the variance parameters $\alpha = 0.01, \gamma = 0.5$. For Q-learning, we set its confidence radius parameter $\beta_k = 0.005$.

**Results** We compare each algorithm with their average accumulated reward, which is the summation of received rewards divided by the number of current episode. We run each algorithm for 500 episodes. Each algorithm has 10 independent runs. We plot their mean values and error bar ([mean-stand error, mean+stand error]) w.r.t. number of episodes. The results for $H = 100$ and $H = 500$ are plotted in Figure 2a and 2b.

---

[3] https://github.com/uclaml/HF-UCRL-VTR

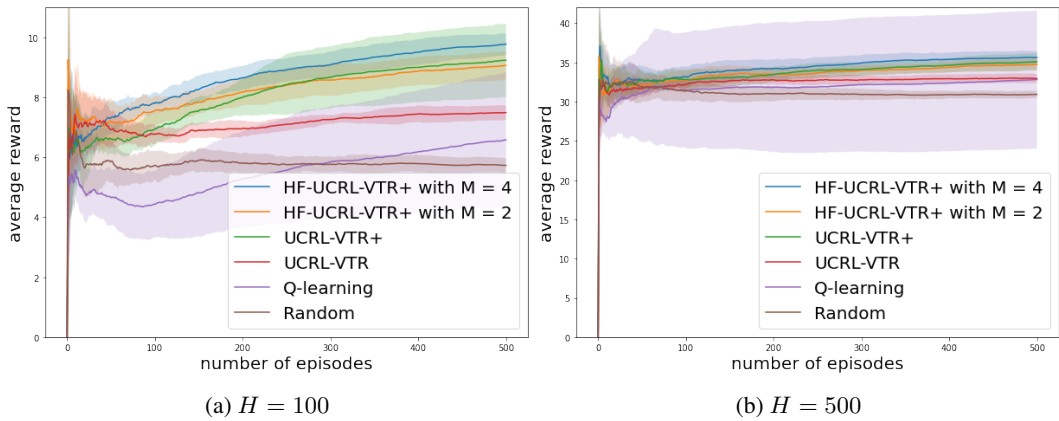

(a) $H = 100$                 (b) $H = 500$

Figure 2: Average accumulated reward of benchmarks.

From Figure 2a and 2b we can draw the following conclusions: (1) HF-UCRL-VTR$^+$ with $M = 4$ outperforms HF-UCRL-VTR$^+$ with $M = 2$, which validates the advantage of the HOME estimator. (2) HF-UCRL-VTR$^+$ and UCRL-VTR$^+$ outperforms UCRL-VTR, which suggests that the Bernstein-type bonus is useful. (3) All UCRL-VTR-based algorithms outperform the tabular Q-learning. The average reward of UCRL-VTR-based algorithms is generally higher than that of Q-learning, and the variance of UCRL-VTR-based algorithms is also smaller than that of Q-learning. This suggests the validity of the value-targeted regression, as well as the use of the feature $\phi$.

## B    Computational complexity of HF-UCRL-VTR$^+$

At each episode $k$ and each stage $h$, HF-UCRL-VTR$^+$ needs to compute $\{\phi_{k,h,m}\}_{m\in\overline{[M]}}$ and $\{\bar{\sigma}_{k,h,m}\}_{m\in\overline{[M]}}$, and update $\{\widetilde{\Sigma}_{k,h+1,m}\}_{m\in\overline{[M]}}$. According to Algorithm 3, $\{\bar{\sigma}_{k,h,m}\}_{m\in\overline{[M]}}$ can be computed in $O(Md^2)$ time as they only require the computation of the inner-product between vectors and the inner-product between an inversion of matrix and an vector. For $\{\phi_{k,h,m}\}_{m\in\overline{[M]}}$, they can be computed within $O(\mathcal{O} \cdot M)$ time. Finally, to selection actions based on $\pi_h^k$, HF-UCRL-VTR$^+$ needs to compute $|\mathcal{A}|$ number of action-value function $Q_{k,h}$, while each of them needs to compute the $\phi_V(\cdot, \cdot)$ within $\mathcal{O}$ time and the inner-product between an inversion of a matrix and a vector by $O(d^2)$ time. Therefore, the total amount of time HF-UCRL-VTR$^+$ takes is $O(KHMd^2 + KH\mathcal{O}M + |\mathcal{A}|KH\mathcal{O} + |\mathcal{A}|KHd^2)$.

**Remark B.1.** Compared with HF-UCRL-VTR$^+$, VOFUL/VOFUL2 (Zhang et al., 2021c; Kim et al., 2021) need to compute the upper bounds of moments as the maximum of the quadratic function $\left[\langle\phi_{k,h,m+1}, \boldsymbol{\theta}\rangle\right]_{[0,1]} - \left[\langle\phi_{k,h,m}, \boldsymbol{\theta}\rangle\right]_{[0,1]}^2$ over a series of implicit confidence sets, which are not implementable.

## C    Proof of the Bernstein-type concentration inequality (Lemma 4.3)

We prove Lemma 4.3 here. First, we provide some new concentration inequalities here, which will be used in the following proof.

**Lemma C.1.** Let $M, v > 0$ be constants. Let $\{x_i\}_{i=1}^n$ be a stochastic process, $\mathcal{G}_i = \sigma(x_1, \dots, x_i)$ be the $\sigma$-algebra of $x_1, \dots, x_i$. Suppose $\mathbb{E}[x_i|\mathcal{G}_{i-1}] = 0$, $|x_i| \leq M$ and $\sum_{i=1}^n \mathbb{E}[x_i^2|\mathcal{G}_{i-1}] \leq v^2$ almost surely. Then for any $\delta, \epsilon > 0$, let $\iota = \sqrt{\log((\log(M/\epsilon) + 2)/\delta)}$, we have

$$\mathbb{P}\left( \sum_{i=1}^n x_i \leq 3\iota v + 9\iota^2 \max_{i\in[n]} |x_i| + 4\iota^2\epsilon \right) > 1 - \delta.$$

*Proof.* For simplicity, let

$$S_n = \sum_{i=1}^n x_i, \ \bar{S}_n = \sqrt{\sum_{i=1}^n \mathbb{E}[x_i^2|\mathcal{G}_{i-1}]}, \ \bar{x}_n = \max_{i \in [n]} |x_i|.$$

We have $\bar{S}_n < v^2, \bar{x}_n \le M$. Set threshold $\epsilon > 0$. Set the following sets

$$\Gamma_0 = [0, \epsilon), \ \Gamma_i = [\epsilon \cdot e^{i-1}, \epsilon \cdot e^i), \ i = 1, \dots, I = \lfloor \log(M/\epsilon) \rfloor + 1.$$

Then there exist $0 \le i \le I$ such that $\bar{x}_n \in \Gamma_i$. Let $\iota = 3\sqrt{\log(1/\delta)}$ and $\zeta = \iota^2$. Taking union bound, we have

$$\mathbb{P}\Big(S_n > \iota v + \zeta \bar{x}_n + \eta\Big) \le \sum_{i=0}^I \mathbb{P}\Big(S_n > \iota v + \zeta \bar{x}_n + \eta, \ \bar{S}_n \le v, \ \bar{x}_n \in \Gamma_i\Big),$$

For each $0 \le i \le I$, we have

$$\mathbb{P}\Big(S_n > \iota v + \zeta \bar{x}_n + \eta, \ \bar{S}_n < v, \ \bar{x}_n \in \Gamma_i\Big)$$

$$\le \mathbb{P}\Big(S_n > \max\{\iota v, \zeta e^{i-1}\epsilon\}, \ \bar{S}_n < v, \ \bar{x}_n < e^i\epsilon\Big)$$

$$\le \mathbb{P}\Big(S_n > \max\{\iota v, \zeta e^{i-1}\epsilon\}, \ \bar{S}_n^2 + \sum_{k=1}^n x_k^2 \mathbb{1}\{|x_k| > e^i\epsilon\} < v^2\Big)$$

$$\le \exp\Big(\frac{-\max\{\iota v, \zeta e^{i-1}\epsilon\}^2}{2(v^2 + e^i\epsilon \cdot \max\{\iota v, \zeta e^{i-1}\epsilon\}/3)}\Big)$$

$$\le \exp\Big(\frac{-1}{2(1/\iota^2 + 1/\zeta)}\Big)$$

$$\le \delta, \tag{C.1}$$

where the first inequality holds since for any $0 \le i \le I$ and $\bar{x}_n \in \Gamma_i$, we have

$$\iota v + \zeta \bar{x}_n + \eta > \iota v + \zeta \bar{x}_n + \zeta \epsilon/e \ge \max\{\iota v, \zeta e^{i-1}\epsilon\},$$

the second one holds since $\bar{x}_n < e^i\epsilon$ implies $\mathbb{1}\{|x_k| > e^j\epsilon\} = 0$ for each $k \in [n]$, the third one holds due to Lemma F.3, the fourth one holds by basic calculation, the last one holds since $\iota = 3\sqrt{\log(1/\delta)}$. Therefore, taking summation for (C.1) over all $0 \le i \le I$, our statement holds. $\square$

**Lemma C.2.** Let $\{x_i \ge 0\}_{i \ge 1}$ be a stochastic process, $\{\mathcal{G}_i\}_{i \ge 1}$ be a filtration satisfying $x_i$ is $\mathcal{G}_i$-measurable. Let $M, v > 0$, and $x_i \le M, \sum_{i=1}^n \mathbb{E}[x_i|\mathcal{G}_{i-1}] \le v$ almost surely. Then for any $\delta, \epsilon > 0$, let $\iota = \log(4(\log(M/\epsilon) + 2)/\delta)$, we have

$$\mathbb{P}\Big(\sum_{i=1}^n x_i \le 4\iota v + 11\iota \max_{1 \le i \le n} x_i + 4\iota\epsilon\Big) > 1 - \delta.$$

*Proof.* For simplicity, let

$$S_n = \sum_{i=1}^n x_i, \ \bar{S}_n = \sum_{i=1}^n \mathbb{E}[x_i|\mathcal{G}_{i-1}], \ \bar{x}_n = \max_{i \in [n]} |x_i|.$$

We have $\bar{S}_n \le v$ and $\bar{x}_n \le M$. Set threshold $\epsilon > 0$. Set the following sets

$$\Gamma_0 = [0, \epsilon), \ \Gamma_i = [\epsilon \cdot e^{i-1}, \epsilon \cdot e^i), \ i = 1, \dots, I = \lfloor \log(M/\epsilon) \rfloor + 1.$$

Then there exist $0 \le i \le I$ such that $\bar{x}_n \in \Gamma_i$. Let $\iota = 4\log(4/\delta), \zeta = 11\log(4/\delta), \eta = 4\log(4/\delta)\epsilon$. By union bound we have

$$\mathbb{P}\Big(S_n > \iota v + \zeta \bar{x}_n + \eta\Big) \le \sum_{i=0}^I \mathbb{P}\Big(S_n > \iota v + \zeta \bar{x}_n + \eta, \ \bar{S}_n \le v, \ \bar{x}_n \in \Gamma_i\Big),$$

For each $0 \leq i \leq I$, we have

$$\mathbb{P}\left(S_n > \iota v + \zeta \bar{x}_n + \eta, \ \bar{S}_n \leq v, \ \bar{x}_n \in \Gamma_i\right)$$

$$\leq \mathbb{P}\left(S_n > \iota v + \zeta \bar{x}_n + \eta, \ \bar{S}_n < \log(4/\delta)(v + e^i\epsilon), \ \bar{x}_n < e^i\epsilon\right)$$

$$= \mathbb{P}\left(\sum_{k=1}^n x_k \mathbb{1}\{x_k \leq e^i\epsilon\} > \iota v + \zeta \bar{x}_n + \eta, \ \bar{S}_n < \log(4/\delta)(v + e^i\epsilon), \ \bar{x}_n < e^i\epsilon\right)$$

$$\leq \mathbb{P}\left(\sum_{k=1}^n x_k \mathbb{1}\{x_k \leq e^i\epsilon\} \geq 4\log(4/\delta)(v + e^i\epsilon), \ \bar{S}_n \leq \log(4/\delta)(v + e^i\epsilon)\right)$$

$$\leq \mathbb{P}\left(\sum_{k=1}^n x_k \mathbb{1}\{x_k \leq e^i\epsilon\} \geq 4\log(4/\delta)(v + e^i\epsilon), \ \sum_{k=1}^n \mathbb{E}[x_k \mathbb{1}\{x_k \leq e^i\epsilon\}|\mathcal{G}_{k-1}] \leq \log(4/\delta)(v + e^i\epsilon)\right)$$

$$= \mathbb{P}\left(\sum_{k=1}^n \frac{x_k \mathbb{1}\{x_k \leq e^i\epsilon\}}{e^i\epsilon} \geq 4\log(4/\delta)\frac{v + e^i\epsilon}{e^i\epsilon}, \ \sum_{k=1}^n \mathbb{E}\left[\frac{x_k \mathbb{1}\{x_k \leq e^i\epsilon\}}{e^i\epsilon}\bigg|\mathcal{G}_{k-1}\right] \leq \log(4/\delta)\frac{v + e^i\epsilon}{e^i\epsilon}\right)$$

$$\leq \delta. \tag{C.2}$$

Here, the second inequality holds since for any $0 \leq i \leq I$ and $\bar{x}_n \in \Gamma_i$, we have

$$\iota v + \zeta \bar{x}_n + \eta = 4\log(4/\delta)v + 11\log(4/\delta)\bar{x}_n + 4\log(4/\delta)\epsilon \geq 4\log(4/\delta)(v + e^i\epsilon).$$

The third inequality holds since $x_k \geq x_k \mathbb{1}\{x_k \leq e^i\epsilon\}$. The last inequality holds by Lemma F.4 with $c = (v + e^i\epsilon)/(e^i\epsilon) > 1$. Therefore, taking summation for (C.2) over all $0 \leq i \leq I$, our statement holds. $\qquad\square$

Now we begin to prove Lemma 4.3. We first define the following notations:

$$\mathbf{d}_1 = \mathbf{0}, \mathbf{d}_k = \sum_{i=1}^k \mathbf{x}_i \eta_i, \ w_k = \|\mathbf{x}_k\|_{\mathbf{Z}_{k-1}^{-1}}, \ \mathcal{E}_k = \left\{1 \leq s \leq k, \|\mathbf{d}_s\|_{\mathbf{Z}_s^{-1}} \leq \beta_s\right\}, \tag{C.3}$$

where $k \geq 1$ and we define $\beta_1 = 0$. Recalling that $x_k$ is $\mathcal{G}_k$-measurable and $\eta_k$ is $\mathcal{G}_{k+1}$-measurable, we find that $\mathbf{d}_k, \mathbf{Z}_k, \mathcal{E}_k$ and $w_k$ is $\mathcal{G}_k$ are $\mathcal{G}_k$-measurable. Recall that $\beta_t$ is defined as follows:

$$\beta_t = 12\sqrt{\sigma^2 d \log(1 + tL^2/(d\lambda)) \log(32(\log(R/\epsilon) + 1)t^2/\delta)}$$
$$+ 24\log(32(\log(R/\epsilon) + 1)t^2/\delta) \max_{1 \leq i \leq t}\{|\eta_i| \min\{1, w_i\}\} + 6\log(32(\log(R/\epsilon) + 1)t^2/\delta)\epsilon.$$

The proof of Lemma 4.3 follows the proof strategy in Dani et al. (2008); Zhou et al. (2021a). Briefly speaking, we can bound the self-normalized martingale $\|\sum_{i=1}^k \mathbf{x}_i \eta_i\|_{\mathbf{Z}_k^{-1}}^2$ by the following two terms,

$$\sum_{i=1}^k \frac{2\eta_i \mathbf{x}_i^\top \mathbf{Z}_{i-1}^{-1} \mathbf{d}_{i-1}}{1 + w_i^2} \mathbb{1}\{\mathcal{E}_{i-1}\}, \text{ and } \sum_{i=1}^k \frac{\eta_i^2 w_i^2}{1 + w_i^2}.$$

Next two lemmas bound these terms by $\beta_t$ separately. The main difference between the following lemmas and their counterparts in Dani et al. (2008); Zhou et al. (2021a) is that our $\beta_t$ is also a *random variable*, which requires us to use some advanced concentration inequalities (Lemmas C.1 and C.2) rather than vanilla ones.

**Lemma C.3.** Let $\mathbf{d}_i, w_i, \mathcal{E}_i$ be defined in (C.3). Then, with probability at least $1 - \delta/2$, simultaneously for all $k \geq 1$ it holds that

$$\sum_{i=1}^k \frac{2\eta_i \mathbf{x}_i^\top \mathbf{Z}_{i-1}^{-1} \mathbf{d}_{i-1}}{1 + w_i^2} \mathbb{1}\{\mathcal{E}_{i-1}\} \leq 3\beta_k^2/4.$$

*Proof.* We have

$$\left|\frac{2\mathbf{x}_i^\top \mathbf{Z}_{i-1}^{-1} \mathbf{d}_{i-1}}{1 + w_i^2} \mathbb{1}\{\mathcal{E}_{i-1}\}\right| \leq \frac{2\|\mathbf{x}_i\|_{\mathbf{Z}_{i-1}^{-1}}[\|\mathbf{d}_{i-1}\|_{\mathbf{Z}_{i-1}^{-1}} \mathbb{1}\{\mathcal{E}_{i-1}\}]}{1 + w_i^2} \leq \frac{2w_i\beta_{i-1}}{1 + w_i^2} \leq \min\{1, 2w_i\}\beta_{i-1}, \tag{C.4}$$

where the first inequality holds due to Cauchy-Schwarz inequality, the second inequality holds due to the definition of $\mathcal{E}_{i-1}$, the last inequality holds by algebra. Fix $t > 0$. For simplicity, let $\ell_i$ denote

$$\ell_i = \frac{2\eta_i \mathbf{x}_i^\top \mathbf{Z}_{i-1}^{-1} \mathbf{d}_{i-1}}{\beta_{i-1}(1 + w_i^2)} \mathbb{1}\{\mathcal{E}_{i-1}\}. \tag{C.5}$$

We are preparing to apply Lemma C.1 to $\{\ell_i\}_i$ and $\{\mathcal{G}_i\}_i$. First note that $\mathbb{E}[\ell_i|\mathcal{G}_i] = 0$. Meanwhile, by (C.4) we have $|\ell_i| \leq R$ and

$$|\ell_i| \leq 2|\eta_i| \min\{1, w_i\}. \tag{C.6}$$

We also have

$$\sum_{i=1}^t \mathbb{E}[\ell_i^2|\mathcal{G}_i] \leq \sigma^2 \sum_{i=1}^t \left( \frac{2\mathbf{x}_i^\top \mathbf{Z}_{i-1}^{-1} \mathbf{d}_{i-1}}{1 + w_i^2} \mathbb{1}\{\mathcal{E}_{i-1}\}/\beta_{i-1} \right)^2$$

$$\leq 4\sigma^2 \sum_{i=1}^t [\min\{1, w_i\}]^2$$

$$\leq 4\sigma^2 \sum_{i=1}^t \min\{1, w_i^2\}$$

$$\leq 8\sigma^2 d \log(1 + tL^2/(d\lambda)), \tag{C.7}$$

where the first inequality holds since $\mathbb{E}[\eta_i^2|\mathcal{G}_i] \leq \sigma^2$, the second inequality holds due to (C.4), the third inequality holds again since $\{\beta_i\}_i$ is increasing, the last inequality holds due to Lemma F.5. Let $\iota_t = \sqrt{\log(4(\log(R/\epsilon) + 2)t^2/\delta)}$. Therefore, by (C.6) and (C.7), using Lemma C.1, we know that with probability at least $1 - \delta/(4t^2)$, we have

$$\sum_{i=1}^t \ell_i \leq 3\iota_t \sqrt{8\sigma^2 d \log(1 + tL^2/(d\lambda))} + 18\iota_t^2 \max_{1 \leq i \leq t}\{|\eta_i| \min\{1, w_i\}\} + 4\iota_t^2 \epsilon \leq \frac{3}{4}\beta_t,$$

where the last inequality holds due to the definition of $\beta_t$. Therefore, using the fact $\beta_{i-1} \leq \beta_t$, we have

$$\sum_{i=1}^t \frac{2\eta_i \mathbf{x}_i^\top \mathbf{Z}_{i-1}^{-1} \mathbf{d}_{i-1}}{(1 + w_i^2)} \mathbb{1}\{\mathcal{E}_{i-1}\} \leq \beta_t \sum_{i=1}^t \ell_i \leq 3\beta_t^2/4. \tag{C.8}$$

Taking union bound for (C.8) from $t = 1$ to $\infty$ and using the fact that $\sum_{t=1}^\infty t^{-2} < 2$ finishes the proof. $\qquad \square$

We also need the following lemma.

**Lemma C.4.** Let $w_i$ be defined in (C.3). Then, with probability at least $1 - \delta/2$, simultaneously for all $k \geq 1$ it holds that

$$\sum_{i=1}^k \frac{\eta_i^2 w_i^2}{1 + w_i^2} \leq \beta_k^2/4.$$

*Proof.* For simplicity, let

$$\ell_i = \frac{\eta_i^2 w_i^2}{1 + w_i^2}.$$

Fix $t$. Then by the fact that $\mathbb{E}[\eta_i^2|\mathcal{G}_i] \leq \sigma^2$ and Lemma F.5, we have

$$\sum_{i=1}^t \mathbb{E}[\ell_i|\mathcal{G}_i] = \sum_{i=1}^t \mathbb{E}\left[\frac{\eta_i^2 w_i^2}{1 + w_i^2}\bigg|\mathcal{G}_i\right] \leq \sigma^2 \sum_{i=1}^t \min\{1, w_i^2\} \leq 2\sigma^2 d \log(1 + tL^2/(d\lambda)). \tag{C.9}$$

Furthermore, we have $|\ell_i| \leq R^2$ and

$$\ell_i \leq |\eta_i|^2 \min\{1, w_i\}^2.$$

Let $\iota_t = \log(32(\log(R/\epsilon) + 1)t^2/\delta)$. Then by Lemma C.2, we have with probability at least $1 - \delta/(4t^2)$,

$$\sum_{i=1}^{t} \ell_i \leq 8\iota_t \sigma^2 d \log(1 + tL^2/(d\lambda)) + 11\iota_t \max_{1 \leq i \leq t} \{|\eta_i| \min\{1, w_i\}\}^2 + 4\iota_t \epsilon^2 \leq \beta_t^2/4. \quad \text{(C.10)}$$

where the last inequality holds due to the definition of $\beta_t$. Taking union bound for (C.10) from $t = 1$ to $\infty$ and using the fact that $\sum_{t=1}^{\infty} t^{-2} < 2$ finishes the proof. $\qquad\square$

With above two lemmas, we are ready to prove Lemma 4.3.

*Proof of Lemma 4.3.* The proof is nearly the same as the proof of Theorem 2 in Zhou et al. (2021a), only to replace Lemmas 13 and 14 in Zhou et al. (2021a) with Lemma C.3 and C.4. $\qquad\square$

# D    Proof of main results in Section 4

In this section, we prove Lemma 4.4 and Theorem 4.1.

## D.1    Proof of Lemma 4.4

We restate Lemma 4.4 here with a different set of notations, as it will be used in the later proof for RL as well.

**Lemma D.1** (Restatement of Lemma 4.4). Let $\{\sigma_k, \beta_k\}_{k \geq 1}$ be a sequence of non-negative numbers, $\alpha, \gamma > 0$, $\{\mathbf{x}_k\}_{k \geq 1} \subset \mathbb{R}^d$ and $\|\mathbf{x}_k\|_2 \leq L$. Let $\{\mathbf{Z}_k\}_{k \geq 1}$ and $\{\bar{\sigma}_k\}_{k \geq 1}$ be recursively defined as follows: $\mathbf{Z}_1 = \lambda \mathbf{I}$,

$$\forall k \geq 1, \ \bar{\sigma}_k = \max\{\sigma_k, \alpha, \gamma \|\mathbf{x}_k\|_{\mathbf{Z}_k^{-1}}^{1/2}\}, \ \mathbf{Z}_{k+1} = \mathbf{Z}_k + \mathbf{x}_k \mathbf{x}_k^\top / \bar{\sigma}_k^2.$$

Let $\iota = \log(1 + KL^2/(d\lambda\alpha^2))$. Then we have

$$\sum_{k=1}^{K} \min\left\{1, \beta_k \|\mathbf{x}_k\|_{\mathbf{Z}_k^{-1}}\right\} \leq 2d\iota + 2\max_{k \in [K]} \beta_k \gamma^2 d\iota + 2\sqrt{d\iota} \sqrt{\sum_{k=1}^{K} \beta_k^2(\sigma_k^2 + \alpha^2)}.$$

*Proof of Lemma D.1.* We decompose the set $[K]$ into a union of two disjoint subsets $[K] = \mathcal{I}_1 \cup \mathcal{I}_2$.

$$\mathcal{I}_1 = \left\{k \in [K] : \|\mathbf{x}_k/\bar{\sigma}_k\|_{\mathbf{Z}_k^{-1}} \geq 1\right\}, \ \mathcal{I}_2 = [K] \setminus \mathcal{I}_1. \quad \text{(D.1)}$$

Then the following upper bound of $|\mathcal{I}_1|$ holds:

$$|\mathcal{I}_1| = \sum_{k \in \mathcal{I}_1} \min\left\{1, \|\mathbf{x}_k/\bar{\sigma}_k\|_{\mathbf{Z}_k^{-1}}^2\right\} \leq \sum_{k=1}^{K} \min\left\{1, \|\mathbf{x}_k/\bar{\sigma}_k\|_{\mathbf{Z}_k^{-1}}^2\right\} \leq 2d\iota, \quad \text{(D.2)}$$

where the first inequality holds since $\|\mathbf{x}_k/\bar{\sigma}_k\|_{\mathbf{Z}_k^{-1}} \geq 1$ for $k \in \mathcal{I}_1$, the third inequality holds due to Lemma F.5 together with the fact $\|\mathbf{x}_k/\bar{\sigma}_k\|_2 \leq L/\alpha$. Therefore, we have

$$\begin{aligned}
&\sum_{k \in [K]} \min\left\{1, \beta_k \|\mathbf{x}_k\|_{\mathbf{Z}_k^{-1}}\right\} \\
&= \sum_{k \in \mathcal{I}_1} \min\left\{1, \bar{\sigma}_k \beta_k \|\mathbf{x}_k/\bar{\sigma}_k\|_{\mathbf{Z}_k^{-1}}\right\} + \sum_{k \in \mathcal{I}_2} \min\left\{1, \bar{\sigma}_k \beta_k \|\mathbf{x}_k/\bar{\sigma}_k\|_{\mathbf{Z}_k^{-1}}\right\} \\
&\leq \left[\sum_{k \in \mathcal{I}_1} 1\right] + \sum_{k \in \mathcal{I}_2} \beta_k \bar{\sigma}_k \|\mathbf{x}_k/\bar{\sigma}_k\|_{\mathbf{Z}_k^{-1}} \\
&\leq 2d\iota + \sum_{k \in \mathcal{I}_2} \beta_k \bar{\sigma}_k \|\mathbf{x}_k/\bar{\sigma}_k\|_{\mathbf{Z}_k^{-1}}, \quad \text{(D.3)}
\end{aligned}$$

where the first inequality holds since $\min\{1, x\} \leq 1$ and also $\min\{1, x\} \leq x$, the second inequality holds due to (D.2). Next we further bound the second summation term in (D.3). We decompose $\mathcal{I}_2 = \mathcal{J}_1 \cup \mathcal{J}_2$, where

$$\mathcal{J}_1 = \left\{ k \in \mathcal{I}_2 : \bar{\sigma}_k = \sigma_k \cup \bar{\sigma}_k = \alpha \right\}, \quad \mathcal{J}_2 = \left\{ k \in \mathcal{I}_2 : \bar{\sigma}_k = \gamma\sqrt{\|\mathbf{x}_k\|_{\mathbf{Z}_k^{-1}}} \right\}.$$

For the summation over $\mathcal{J}_1$, we have

$$\sum_{k \in \mathcal{J}_1} \beta_k \bar{\sigma}_k \|\mathbf{x}_k/\bar{\sigma}_k\|_{\mathbf{Z}_k^{-1}} \leq \sum_{k \in \mathcal{J}_1} \beta_k(\sigma_k + \alpha) \min\{1, \|\mathbf{x}_k/\bar{\sigma}_k\|_{\mathbf{Z}_k^{-1}}\}$$

$$\leq \sum_{k=1}^{K} \beta_k(\sigma_k + \alpha)\min\{1, \|\mathbf{x}_k/\bar{\sigma}_k\|_{\mathbf{Z}_k^{-1}}\}$$

$$\leq \sqrt{2\sum_{k=1}^{K}(\sigma_k^2 + \alpha^2)\beta_k^2}\sqrt{\sum_{k=1}^{K}\min\{1, \|\mathbf{x}_k/\bar{\sigma}_k\|_{\mathbf{Z}_k^{-1}}\}^2}$$

$$\leq 2\sqrt{\sum_{k=1}^{K}\beta_k^2(\sigma_k^2 + \alpha^2)}\sqrt{d\iota}, \tag{D.4}$$

where the first inequality holds since $\bar{\sigma}_k \leq \sigma_k + \alpha$ for $k \in \mathcal{J}_1$ and $\|\mathbf{x}_k/\bar{\sigma}_k\|_{\mathbf{Z}_k^{-1}} \leq 1$ since $k \in \mathcal{J}_1 \subseteq \mathcal{I}_2$, the third one holds due to Cauchy-Schwarz inequality and the last one holds due to Lemma F.5. Next we bound the summation over $\mathcal{J}_2$. First, for $k \in \mathcal{J}_2$, we have the following equation:

$$\bar{\sigma}_k = \gamma^2\|\mathbf{x}_k/\bar{\sigma}_k\|_{\mathbf{Z}_k^{-1}}.$$

Then we can bound the summation over $\mathcal{J}_2$ as follows:

$$\sum_{k \in \mathcal{J}_2} \beta_k \bar{\sigma}_k \|\mathbf{x}_k/\bar{\sigma}_k\|_{\mathbf{Z}_k^{-1}} = \gamma^2 \cdot \sum_{k \in \mathcal{J}_1} \beta_k \|\mathbf{x}_k/\bar{\sigma}_k\|_{\mathbf{Z}_k^{-1}}^2 = \gamma^2 \cdot \sum_{k=1}^{K} \beta_k \min\{1, \|\mathbf{x}_k/\bar{\sigma}_k\|_{\mathbf{Z}_k^{-1}}^2\} \leq 2\max_{k \in [K]} \beta_k \gamma^2 d\iota, \tag{D.5}$$

where the inequality holds due to Lemma F.5. Substituting (D.4) and (D.5) into (D.3) completes the proof. $\qquad\square$

## D.2 Proof of Theorem 4.1

*Proof of Theorem 4.1.* By the assumption on $\epsilon_k$, we know that

$$|\epsilon_k/\bar{\sigma}_k| \leq R/\alpha,$$

$$|\epsilon_k/\bar{\sigma}_k| \cdot \min\{1, \|\mathbf{a}_k/\bar{\sigma}_k\|_{\widehat{\mathbf{\Sigma}}_k^{-1}}\} \leq R\|\mathbf{a}_k\|_{\widehat{\mathbf{\Sigma}}_k^{-1}}/\bar{\sigma}_k^2 \leq R/\gamma^2,$$

$$\mathbb{E}[\epsilon_k|\mathbf{a}_{1:k}, \epsilon_{1:k-1}] = 0, \quad \mathbb{E}[(\epsilon_k/\bar{\sigma}_k)^2|\mathbf{a}_{1:k}, \epsilon_{1:k-1}] \leq 1, \quad \|\mathbf{a}_k/\bar{\sigma}_k\|_2 \leq A/\alpha,$$

Therefore, setting $\mathcal{G}_k = \sigma(\mathbf{a}_{1:k}, \epsilon_{1:k-1})$, and using that $\sigma_k$ is $\mathcal{G}_k$-measurable, applying Theorem 4.3 to $(\boldsymbol{x}_k, \eta_k) = (\boldsymbol{a}_k/\bar{\sigma}_k, \epsilon_k/\bar{\sigma}_k)$ with $\epsilon = R/\gamma^2$, we get that with probability at least $1 - \delta$, for all $k \geq 1$,

$$\|\widehat{\boldsymbol{\theta}}_k - \boldsymbol{\theta}^*\|_{\widehat{\mathbf{\Sigma}}_k} \leq 12\sqrt{d\log(1 + kA^2/(\alpha^2 d\lambda))\log(32(\log(\gamma^2/\alpha) + 1)k^2/\delta)}$$

$$+ 30\log(32(\log(\gamma^2/\alpha) + 1)k^2/\delta)R/\gamma^2 + \sqrt{\lambda}B$$

$$:= \widehat{\beta}_k, \tag{D.6}$$

Let $\mathcal{E}_{D.6}$ denote the event such that (D.6) happens for all $k \geq 1$. Then, on the event $\mathcal{E}_{D.6}$, we have

$$\langle \mathbf{a}_k^*, \boldsymbol{\theta}^* \rangle \leq \langle \mathbf{a}_k^*, \widehat{\boldsymbol{\theta}}_k \rangle + \|\mathbf{a}_k^*\|_{\widehat{\mathbf{\Sigma}}_k^{-1}}\|\widehat{\boldsymbol{\theta}}_k - \boldsymbol{\theta}^*\|_{\widehat{\mathbf{\Sigma}}_k} \leq \langle \mathbf{a}_k^*, \widehat{\boldsymbol{\theta}}_k \rangle + \widehat{\beta}_k\|\mathbf{a}_k^*\|_{\widehat{\mathbf{\Sigma}}_k^{-1}} \leq \langle \mathbf{a}_k, \widehat{\boldsymbol{\theta}}_k \rangle + \widehat{\beta}_k\|\mathbf{a}_k\|_{\widehat{\mathbf{\Sigma}}_k^{-1}}, \tag{D.7}$$

where the last inequality holds due to the selection of $\mathbf{a}_k$. By (D.7), we have

$$\langle \mathbf{a}_k^*, \boldsymbol{\theta}^* \rangle - \langle \mathbf{a}_k, \boldsymbol{\theta}^* \rangle \leq \|\mathbf{a}_k\|_{\widehat{\mathbf{\Sigma}}_k^{-1}}(\widehat{\beta}_k + \|\boldsymbol{\theta}^* - \widehat{\boldsymbol{\theta}}_k\|_{\widehat{\mathbf{\Sigma}}_k}) \leq 2\widehat{\beta}_k\|\mathbf{a}_k\|_{\widehat{\mathbf{\Sigma}}_k^{-1}}, \tag{D.8}$$

where the first inequality holds due to Cauchy-Schwarz inequality, the second one holds due to event $\mathcal{E}_{D.6}$. Meanwhile, we have $0 \leq \langle \mathbf{a}_k^*, \boldsymbol{\theta}^* \rangle - \langle \mathbf{a}_k, \boldsymbol{\theta}^* \rangle \leq 2$. Thus, substituting (D.8) into (D.7) and summing up (D.7) for $k \in [K]$, we have

$$\text{Regret}(K) = \sum_{k=1}^{K} \left[ \langle \mathbf{a}_k^*, \boldsymbol{\theta}^* \rangle - \langle \mathbf{a}_k, \boldsymbol{\theta}^* \rangle \right] \leq 2 \sum_{k=1}^{K} \min \left\{ 1, \widehat{\beta}_K \|\mathbf{a}_k\|_{\widehat{\boldsymbol{\Sigma}}_k^{-1}} \right\}. \tag{D.9}$$

Finally, to bound (D.9), by Lemma D.1 we have

$$\text{Regret}(K)/2 \leq 2d\iota + 2d\gamma^2 \widehat{\beta}_K \iota + \widehat{\beta}_K \sqrt{2 \sum_{k=1}^{K} \sigma_k^2 + 2K\alpha^2} \sqrt{2d\iota},$$

which completes our proof. $\qquad\qquad\square$

# E  Proof of main results in Section 5

For $k \in [K]$, $h \in [H]$, let $\mathcal{F}_{k,h}$ be the $\sigma$-algebra generated by the random variables representing the state-action pairs up to and including those that appear stage $h$ of episode $k$. That is, $\mathcal{F}_{k,h}$ is generated by

$$s_1^1, a_1^1, \ldots, s_h^1, a_h^1, \ldots, s_H^1, a_H^1,$$
$$s_1^2, a_1^2, \ldots, s_h^2, a_h^2, \ldots, s_H^2, a_H^2,$$
$$\vdots$$
$$s_1^k, a_1^k, \ldots, s_h^k, a_h^k.$$

Note that, by construction,

$$\bar{\mathbb{V}}_{k,h} V_{k,h+1}^{2^m}(s_h^k, a_h^k), E_{k,h,m}, \bar{\sigma}_{k,h,m}, \widetilde{\boldsymbol{\Sigma}}_{k,h+1,m}, \phi_{k,h,m}$$

are $\mathcal{F}_{k,h}$-measurable, and $Q_{k,h}, V_{k,h}, \pi_h^k, \widehat{\boldsymbol{\Sigma}}_{k,m}, \widehat{\boldsymbol{b}}_{k,m}, \widehat{\boldsymbol{\theta}}_{k,m}$ are $\mathcal{F}_{k-1,H}$ measurable.

First we propose a lemma that suggests that the vector $\boldsymbol{\theta}^*$ lies in a series of confidence sets, which further implies that the difference between the estimated high-order moment and true moment can be bounded by error terms $E_{k,h,m}$.

**Lemma E.1.** Set $\{\widehat{\beta}_k\}_{k \geq 1}$ as (5.3), then, with probability at least $1 - M\delta$, we have for any $k \in [K], h \in [H], m \in \overline{[M]}$,

$$\left\| \widehat{\boldsymbol{\Sigma}}_{k,m}^{1/2} \left( \widehat{\boldsymbol{\theta}}_{k,m} - \boldsymbol{\theta}^* \right) \right\|_2 \leq \widehat{\beta}_k, \; |[\bar{\mathbb{V}}_{k,m} V_{k,h+1}^{2^m}](s_h^k, a_h^k) - [\mathbb{V} V_{k,h+1}^{2^m}](s_h^k, a_h^k)| \leq E_{k,h,m}.$$

## E.1  Proof of Lemma E.1

We start with the following lemma.

**Lemma E.2.** Let $V_{k,h}, \widehat{\boldsymbol{\Sigma}}_{k,m}, \widehat{\boldsymbol{\theta}}_{k,m}, \phi_{k,h,m}, \bar{\mathbb{V}}_{k,m}$ be defined in Algorithm 2. For any $k \in [K], h \in [H], m \in \overline{[M]}$, we have

$$\left| \mathbb{V} V_{k,h+1}^{2^m}(s_h^k, a_h^k) - \bar{\mathbb{V}}_{k,m} V_{k,h+1}^{2^m}(s_h^k, a_h^k) \right|$$
$$\leq \min \left\{ 1, \left\| \widehat{\boldsymbol{\Sigma}}_{k,m+1}^{-1/2} \phi_{k,h,m+1} \right\|_2 \left\| \widehat{\boldsymbol{\Sigma}}_{k,m+1}^{1/2} \left( \widehat{\boldsymbol{\theta}}_{k,m+1} - \boldsymbol{\theta}^* \right) \right\|_2 \right\}$$
$$+ \min \left\{ 1, 2 \left\| \widehat{\boldsymbol{\Sigma}}_{k,m}^{-1/2} \phi_{k,h,m} \right\|_2 \left\| \widehat{\boldsymbol{\Sigma}}_{k,m}^{1/2} \left( \widehat{\boldsymbol{\theta}}_{k,m} - \boldsymbol{\theta}^* \right) \right\|_2 \right\}.$$

*Proof.* The proof is nearly identical to the proof of Lemma 15 in Zhou et al. (2021a). The only difference is to replace the upper bound $H$ with 1, $V_{k,h+1}$ with $V_{k,h+1}^{2^m}$, $\widetilde{\boldsymbol{\Sigma}}_{k,h}$ with $\widehat{\boldsymbol{\Sigma}}_{k,m+1}$, $\widetilde{\boldsymbol{\theta}}_{k,h}$ with $\widehat{\boldsymbol{\theta}}_{k,m+1}$, $\widehat{\boldsymbol{\Sigma}}_{k,h}$ with $\widehat{\boldsymbol{\Sigma}}_{k,m}$, $\widehat{\boldsymbol{\theta}}_{k,h}$ with $\widehat{\boldsymbol{\theta}}_{k,m}$, and $\boldsymbol{\theta}_h^*$ with $\boldsymbol{\theta}^*$. $\qquad\square$

*Proof of Lemma E.1.* First we recall the definition of $\bar{\sigma}_{k,h,m}$:

$$\bar{\sigma}_{k,h,m}^2 = \max\left\{[\bar{\mathbb{V}}_{k,m}V_{k,h+1}^{2^m}](s_h^k, a_h^k) + E_{k,h,m}, \alpha^2, \gamma^2\left\|\widehat{\boldsymbol{\Sigma}}_{k,m}^{-1/2}\boldsymbol{\phi}_{k,h,m}\right\|_2\right\},$$

$$\bar{\sigma}_{k,h,M-1}^2 = \max\left\{1, \alpha^2, \gamma^2\left\|\widehat{\boldsymbol{\Sigma}}_{k,M-1}^{-1/2}\boldsymbol{\phi}_{k,h,M-1}\right\|_2\right\},$$

We prove the statement by induction. We define the following quantities. For simplicity, let $\widehat{\mathcal{C}}_{k,m}$ be defined as

$$\widehat{\mathcal{C}}_{k,m} := \left\{\boldsymbol{\theta} : \left\|\widehat{\boldsymbol{\Sigma}}_{k,m}^{1/2}(\widehat{\boldsymbol{\theta}}_{k,m} - \boldsymbol{\theta})\right\|_2 \le \widehat{\beta}_k\right\}.$$

For each $m$, let

$$\mathbf{x}_{k,h,m} = \bar{\sigma}_{k,h,m}^{-1}\boldsymbol{\phi}_{k,h,m},$$

$$\eta_{k,h,m} = \bar{\sigma}_{k,h,m}^{-1} \mathbb{1}\{\boldsymbol{\theta}^* \in \widehat{\mathcal{C}}_{k,m} \cap \widehat{\mathcal{C}}_{k,m+1}\}\left[V_{k,h+1}^{2^m}(s_{h+1}^k) - \langle\boldsymbol{\phi}_{k,h,m}, \boldsymbol{\theta}^*\rangle\right],$$

$$\eta_{k,h,M-1} = \bar{\sigma}_{k,h,M-1}^{-1}\left[V_{k,h+1}^{2^{M-1}}(s_{h+1}^k) - \langle\boldsymbol{\phi}_{k,h,M-1}, \boldsymbol{\theta}^*\rangle\right]$$

$$\mathcal{G}_{k,h} = \mathcal{F}_{k,h},$$

$$\boldsymbol{\mu}^* = \boldsymbol{\theta}^*.$$

We have

$$\mathbb{E}[\eta_{k,h,m}|\mathcal{G}_{k,h}] = 0, \ \|\mathbf{x}_{k,h,m}\|_2 \le \bar{\sigma}_{k,h,m}^{-1} \le 1/\alpha, \ |\eta_{k,h,m}| \le 1/\alpha.$$

Note that $\mathbb{1}\{\boldsymbol{\theta}^* \in \widehat{\mathcal{C}}_{k,m} \cap \widehat{\mathcal{C}}_{k,m+1}\}$ is $\mathcal{G}_{k,h}$-measurable, then we have bound the variance of $\eta_{k,h,m}$ as follows: for $m \in \overline{[M-1]}$,

$$\mathbb{E}[\eta_{k,h,m}^2|\mathcal{G}_{k,h}] = \bar{\sigma}_{k,h,m}^{-2} \mathbb{1}\{\boldsymbol{\theta}^* \in \widehat{\mathcal{C}}_{k,m} \cap \widehat{\mathcal{C}}_{k,m+1}\}[\mathbb{V}V_{k,h+1}^{2^m}](s_h^k, a_h^k)$$

$$\le \bar{\sigma}_{k,h,m}^{-2} \mathbb{1}\{\boldsymbol{\theta}^* \in \widehat{\mathcal{C}}_{k,m} \cap \widehat{\mathcal{C}}_{k,m+1}\}\left[\bar{\mathbb{V}}_{k,m}V_{k,h+1}^{2^m}(s_h^k, a_h^k)\right.$$

$$+ \min\left\{1, \left\|\widehat{\boldsymbol{\Sigma}}_{k,m+1}^{-1/2}\boldsymbol{\phi}_{k,h,m+1}\right\|_2\left\|\widehat{\boldsymbol{\Sigma}}_{k,m+1}^{1/2}(\widehat{\boldsymbol{\theta}}_{k,m+1} - \boldsymbol{\theta}^*)\right\|_2\right\}$$

$$+ \min\left\{1, 2\left\|\widehat{\boldsymbol{\Sigma}}_{k,m}^{-1/2}\boldsymbol{\phi}_{k,h,m}\right\|_2\left\|\widehat{\boldsymbol{\Sigma}}_{k,m}^{1/2}(\widehat{\boldsymbol{\theta}}_{k,m} - \boldsymbol{\theta}^*)\right\|_2\right\}\right]$$

$$\le \bar{\sigma}_{k,h,m}^{-2}\left[\bar{\mathbb{V}}_{k,m}V_{k,h+1}^{2^m}(s_h^k, a_h^k) + \min\left\{1, \widehat{\beta}_k\left\|\widehat{\boldsymbol{\Sigma}}_{k,m+1}^{-1/2}\boldsymbol{\phi}_{k,h,m+1}\right\|_2\right\}\right.$$

$$+ \min\left\{1, 2\widehat{\beta}_k\left\|\widehat{\boldsymbol{\Sigma}}_{k,m}^{-1/2}\boldsymbol{\phi}_{k,h,m}\right\|_2\right\}\right]$$

$$\le 1.$$

For $m = M - 1$, we directly have $\mathbb{E}[\eta_{k,h,M-1}^2|\mathcal{G}_{k,h}] \le 1$. Meanwhile, for any $m \in \overline{[M]}$, we have

$$|\eta_{k,h,m}|\max\{1, \|\mathbf{x}_{k,h,m}\|_{\widetilde{\boldsymbol{\Sigma}}_{k,h-1,m}^{-1}}\} \le \bar{\sigma}_{k,h,m}^{-2}\|\boldsymbol{\phi}_{k,h,m}\|_{\widetilde{\boldsymbol{\Sigma}}_{k,h-1,m}^{-1}} \le 1/\gamma^2.$$

Now, let $y, \mathbf{Z}, \mathbf{b}, \boldsymbol{\mu}, \epsilon$ defined in Theorem 4.3 be set as follows:

$$y_{k,h,m} = \langle\boldsymbol{\mu}^*, \mathbf{x}_{k,h,m}\rangle + \eta_{k,h,m}$$

$$\mathbf{Z}_{k,m} = \lambda\mathbf{I} + \sum_{i=1}^{k}\sum_{h=1}^{H}\mathbf{x}_{i,h,m}\mathbf{x}_{i,h,m}^\top = \widehat{\boldsymbol{\Sigma}}_{k+1,m}$$

$$\mathbf{b}_{k,m} = \sum_{i=1}^{k}\sum_{h=1}^{H}\mathbf{x}_{i,h,m}y_{i,h,m}$$

$$\boldsymbol{\mu}_{k,m} = \mathbf{Z}_{k,m}^{-1}\mathbf{b}_{k,m},$$

$$\epsilon = 1/\gamma^2.$$

Then, by Theorem 4.3, for each $m \in \overline{[M]}$, with probability at least $1 - \delta$, $\forall k \in [K+1]$,

$$\|\boldsymbol{\mu}_{k-1,m} - \boldsymbol{\theta}^*\|_{\widehat{\boldsymbol{\Sigma}}_{k,m}} \leq 12\sqrt{d\log(1+kH/(\alpha^2 d\lambda))\log(32(\log(\gamma^2/\alpha)+1)k^2 H^2/\delta)}$$

$$+ 30\log(32(\log(\gamma^2/\alpha)+1)k^2 H^2/\delta)/\gamma^2 + \sqrt{\lambda}B$$

$$= \widehat{\beta}_k. \tag{E.1}$$

Denote the event that (E.1) happens for all $k$ and $m$ as $\mathcal{E}$. Conditioned on $\mathcal{E}$, we have the following observations:

- For $k = 1$, $m \in \overline{[M]}$, by the definitions of $\widehat{\boldsymbol{\theta}}_{1,m}, \widehat{\boldsymbol{\Sigma}}_{1,m}$, we have $\|\boldsymbol{\theta}^* - \widehat{\boldsymbol{\theta}}_{1,m}\|_{\widehat{\boldsymbol{\Sigma}}_{1,m}} = \|\boldsymbol{\theta}^*\|_{\lambda \mathbf{I}} \leq \sqrt{\lambda}B = \widehat{\beta}_1$, which implies

$$\boldsymbol{\theta}^* \in \widehat{\mathcal{C}}_{1,m}. \tag{E.2}$$

- For $k \in [K]$ and $m = M - 1$, we directly have $\boldsymbol{\mu}_{k,M-1} = \widehat{\boldsymbol{\theta}}_{k+1,M-1}$, which implies

$$\boldsymbol{\theta}^* \in \widehat{\mathcal{C}}_{k+1,M-1}. \tag{E.3}$$

- For $k \in [K]$ and $m \in \overline{[M-1]}$, we have

$$\boldsymbol{\theta}^* \in \widehat{\mathcal{C}}_{k,m} \cap \widehat{\mathcal{C}}_{k,m+1} \Rightarrow y_{k,h,m} = \bar{\sigma}_{k,h,m}^{-1} V_{k,h+1}^{2^m}(s_{h+1}^k) \Rightarrow \boldsymbol{\mu}_{k,m} = \widehat{\boldsymbol{\theta}}_{k+1,m} \Rightarrow \boldsymbol{\theta}^* \in \widehat{\mathcal{C}}_{k+1,m}. \tag{E.4}$$

Therefore by induction based on the initial conditions (E.2) and (E.3) and induction rule (E.4), we have for $k \in [K]$ and $m \in \overline{[M]}$, $\boldsymbol{\theta}^* \in \widehat{\mathcal{C}}_{k,m}$. Lastly, conditioned on $\mathcal{E}$, by Lemma E.1, we have for all $k \in [K], h \in [H], m \in \overline{[M]}$,

$$\left|[\bar{\mathbb{V}}_{k,m}V_{k,h+1}^{2^m}](s_h^k, a_h^k) - [\mathbb{V}V_{k,h+1}^{2^m}](s_h^k, a_h^k)\right| \leq E_{k,h,m}.$$

$\square$

## E.2  Proof of Theorem 5.1

Let $\mathcal{E}_{E.1}$ denote the event described by Lemma E.1. We have the following lemmas.

**Lemma E.3.** Let $Q_{k,h}, V_{k,h}$ be defined in Algorithm 2. Then, on the event $\mathcal{E}_{E.1}$, for any $s, a, k, h$ we have that $Q_h^*(s,a) \leq Q_{k,h}(s,a)$, $V_h^*(s) \leq V_{k,h}(s)$.

*Proof.* The proof is nearly identical to the proof of Lemma 19 in Zhou et al. 2021a. The only difference is to replace the event used in Lemma 19 in Zhou et al. (2021a) with the event defined by Lemma E.1. Besides, we replace $\widehat{\mathcal{C}}_{k,h}$ with $\widehat{\mathcal{C}}_{k,0}$, $\widehat{\boldsymbol{\theta}}_{k,h}$ with $\widehat{\boldsymbol{\theta}}_{k,0}$, $\widehat{\boldsymbol{\Sigma}}_{k,h}$ with $\widehat{\boldsymbol{\Sigma}}_{k,0}$, $\mathbb{P}_h$ with $\mathbb{P}$ and $\boldsymbol{\theta}_h^*$ with $\boldsymbol{\theta}$. $\square$

**Lemma E.4.** Let $V_{k,h}, \widehat{\boldsymbol{\Sigma}}_{k,0}, \boldsymbol{\phi}_{k,h,0}$ be defined in Algorithm 2, $\widehat{\beta}_k$ be defined in (5.3). Then on the event $\mathcal{E}_{E.1}$, for any $k \in [K], h \in [H]$, we have

$$V_{k,h}(s_h^k) - r(s_h^k, a_h^k) - [\mathbb{P}V_{k,h+1}](s_h^k, a_h^k) \leq 2\min\{1, \widehat{\beta}_k\left\|\widehat{\boldsymbol{\Sigma}}_{k,0}^{-1/2}\boldsymbol{\phi}_{k,h,0}\right\|_2\} \tag{E.5}$$

*Proof.* For any $k \in [K]$ and $h \in [H]$, we have

$$V_{k,h}(s_h^k) - r(s_h^k, a_h^k) - [\mathbb{P}V_{k,h+1}](s_h^k, a_h^k)$$

$$\leq \langle \boldsymbol{\phi}_{V_{k,h+1}}(s_h^k, a_h^k), \widehat{\boldsymbol{\theta}}_{k,0}\rangle + \widehat{\beta}_k\left\|\widehat{\boldsymbol{\Sigma}}_{k,0}^{-1/2}\boldsymbol{\phi}_{V_{k,h+1}}(s_h^k, a_h^k)\right\|_2 - \langle \boldsymbol{\phi}_{V_{k,h+1}}(s_h^k, a_h^k), \boldsymbol{\theta}^*\rangle$$

$$\leq \left\|\widehat{\boldsymbol{\Sigma}}_{k,0}^{1/2}(\widehat{\boldsymbol{\theta}}_{k,0} - \boldsymbol{\theta}^*)\right\|_2\left\|\widehat{\boldsymbol{\Sigma}}_{k,0}^{-1/2}\boldsymbol{\phi}_{k,h,0}\right\|_2 + \widehat{\beta}_k\left\|\widehat{\boldsymbol{\Sigma}}_{k,0}^{-1/2}\boldsymbol{\phi}_{k,h,0}\right\|_2$$

$$\leq 2\widehat{\beta}_k\left\|\widehat{\boldsymbol{\Sigma}}_{k,0}^{-1/2}\boldsymbol{\phi}_{k,h,0}\right\|_2, \tag{E.6}$$

where the first inequality holds due to the definition of $V_{k,h}$, the second one holds due to Cauchy-Schwarz inequality and the last one holds since on $\mathcal{E}_{E.1}$, $\widehat{\boldsymbol{\theta}}_{k,0} \in \widehat{\mathcal{C}}_{k,0}$. Meanwhile, we have $V_{k,h}(s_h^k) - r(s_h^k, a_h^k) - [\mathbb{P}V_{k,h+1}](s_h^k, a_h^k) \leq 2$. Therefore, combining two bounds completes the proof. $\square$

Recall $\iota = \log(1 + KH/(d\lambda\alpha^2))$, $\zeta = 4\log(4\log(KH)/\delta)$. For any $k \in [K], h \in [H]$ we define the following indicator function $I_h^k$ such that

$$I_h^k := \mathbb{1}\left\{\forall m \in \overline{[M]},\ \det(\widehat{\boldsymbol{\Sigma}}_{k,m}^{-1/2})/\det(\widetilde{\boldsymbol{\Sigma}}_{k,h,m}^{-1/2}) \leq 4\right\}.$$

Clearly $I_h^k$ is $\mathcal{F}_{k,h}$-measurable and monotonically decreasing. For $m \in \overline{[M]}$, we also define the following quantities:

$$R_m = \sum_{k=1}^{K}\sum_{h=1}^{H} I_h^k \min\left\{1, \widehat{\beta}_k\left\|\widehat{\boldsymbol{\Sigma}}_{k,m}^{-1/2}\phi_{k,h,m}\right\|_2\right\}, \tag{E.7}$$

$$A_m = \sum_{k=1}^{K}\sum_{h=1}^{H} I_h^k[[\mathbb{P}V_{k,h+1}^{2^m}](s_h^k, a_h^k) - V_{k,h+1}^{2^m}(s_{h+1}^k)], \tag{E.8}$$

$$S_m = \sum_{k=1}^{K}\sum_{h=1}^{H} I_h^k[\mathbb{V}V_{k,h+1}^{2^m}](s_h^k, a_h^k), \tag{E.9}$$

$$G = \sum_{k=1}^{K}(1 - I_H^k). \tag{E.10}$$

We aim to bound $R_m, A_m, S_m, G$ by the following lemmas.

**Lemma E.5.** *Let $\gamma, \alpha$ be defined in Algorithm 2, $\{R_m, S_m\}_{m\in\overline{[M]}}$ be defined in (E.7) and (E.9). Then for $m \in \overline{[M-1]}$, we have*

$$R_m \leq \min\{4d\iota + 4\widehat{\beta}_K\gamma^2 d\iota + 2\widehat{\beta}_K\sqrt{d\iota}\sqrt{S_m + 4R_m + 2R_{m+1} + KH\alpha^2}, KH\}. \tag{E.11}$$

*For $R_{M-1}$, we have $R_{M-1} \leq KH$.*

*Proof.* For $(k, h)$ where $I_h^k = 1$, by Lemma F.6 we have

$$\left\|\widehat{\boldsymbol{\Sigma}}_{k,m}^{-1/2}\phi_{k,h,m}\right\|_2 \leq \left\|\widetilde{\boldsymbol{\Sigma}}_{k,h,m}^{-1/2}\phi_{k,h,m}\right\|_2 \cdot \sqrt{\frac{\det\widehat{\boldsymbol{\Sigma}}_{k,m}^{-1/2}}{\det\widetilde{\boldsymbol{\Sigma}}_{k,h,m}^{-1/2}}} \leq 2\left\|\widetilde{\boldsymbol{\Sigma}}_{k,h,m}^{-1/2}\phi_{k,h,m}\right\|_2, \tag{E.12}$$

then substituting (E.12) into (E.7), we have

$$R_m \leq 2\sum_{k=1}^{K}\sum_{h=1}^{H}\min\{1, I_h^k\widehat{\beta}_k\left\|\widetilde{\boldsymbol{\Sigma}}_{k,h,m}^{-1/2}\phi_{k,h,m}\right\|_2\}. \tag{E.13}$$

(E.13) can be bounded by Lemma D.1, with $\beta_{k,h} = I_h^k\widehat{\beta}_k$, $\bar{\sigma}_{k,h} = \bar{\sigma}_{k,h,m}$, $\mathbf{a}_{k,h} = \phi_{k,h,m}$ and $\widehat{\boldsymbol{\Sigma}}_{k,h} = \widetilde{\boldsymbol{\Sigma}}_{k,h,m}$. We have

$$\sum_{k=1}^{K}\sum_{h=1}^{H}\min\{1, I_h^k\widehat{\beta}_k\left\|\widetilde{\boldsymbol{\Sigma}}_{k,h,m}^{-1/2}\phi_{k,h,m}\right\|_2\}$$

$$\leq 2d\iota + 2\widehat{\beta}_K\gamma^2 d\iota + 2\widehat{\beta}_K\sqrt{\sum_{k=1}^{K}\sum_{h=1}^{H} I_h^k\left[[\bar{\mathbb{V}}_{k,m}V_{k,h+1}^{2^m}](s_h^k, a_h^k) + E_{k,h,m}\right] + KH\alpha^2}\sqrt{d\iota}$$

$$\leq 2d\iota + 2\widehat{\beta}_K\gamma^2 d\iota + 2\widehat{\beta}_K\sqrt{\sum_{k=1}^{K}\sum_{h=1}^{H} I_h^k\left[[\mathbb{V}V_{k,h+1}^{2^m}](s_h^k, a_h^k) + 2E_{k,h,m}\right] + KH\alpha^2}\sqrt{d\iota}$$

Note that

$$\sum_{k=1}^{K}\sum_{h=1}^{H} I_h^k E_{k,h,m} = \sum_{k=1}^{K}\sum_{h=1}^{H} I_h^k \min\left\{1, 2\widehat{\beta}_k\left\|\widehat{\boldsymbol{\Sigma}}_{k,m}^{-1/2}\phi_{k,h,m}\right\|_2\right\}$$

$$+ \sum_{k=1}^{K} \sum_{h=1}^{H} I_h^k \min\left\{1, \widehat{\beta}_k \left\|\widehat{\mathbf{\Sigma}}_{k,m+1}^{-1/2} \phi_{k,h,m+1}\right\|_2\right\}$$
$$\leq 2R_m + R_{m+1},$$

by the definition of $R_m$ in (E.7). This completes our proof. $\qquad\square$

**Lemma E.6** (Lemma 25, Zhang et al. 2021c). *Let* $\{S_m, A_m\}_{m \in \overline{[M]}}$ *be defined in (E.8) and (E.9), $G$ be defined in (E.10). Then on the event $\mathcal{E}_{E.1}$, for $m \in \overline{[M-1]}$, we have*

$$S_m \leq |A_{m+1}| + G + 2^{m+1}(K + 2R_0). \tag{E.14}$$

*Proof.* The proof follows the proof of Lemma 25 in Zhang et al. 2021c. We have

$$S_m = \sum_{k=1}^{K} \sum_{h=1}^{H} I_h^k [\mathbb{P}V_{k,h+1}^{2^{m+1}}](s_h^k, a_h^k) - ([\mathbb{P}V_{k,h+1}^{2^m}](s_h^k, a_h^k))^2]$$

$$= \sum_{k=1}^{K} \sum_{h=1}^{H} I_h^k [\mathbb{P}V_{k,h+1}^{2^{m+1}}](s_h^k, a_h^k) - V_{k,h+1}^{2^{m+1}}(s_{h+1}^k)] + I_h^k [V_{k,h}^{2^{m+1}}(s_h^k) - ([\mathbb{P}V_{k,h+1}^{2^m}](s_h^k, a_h^k))^2]$$

$$+ \sum_{k=1}^{K} \sum_{h=1}^{H} I_h^k (V_{k,h+1}^{2^{m+1}}(s_{h+1}^k) - V_{k,h}^{2^{m+1}}(s_h^k))$$

$$\leq A_{m+1} + \sum_{k=1}^{K} \sum_{h=1}^{H} I_h^k [V_{k,h}^{2^{m+1}}(s_h^k) - ([\mathbb{P}V_{k,h+1}^{2^m}](s_h^k, a_h^k))^2] + \sum_{k=1}^{K} I_{h_k}^k V_{k,h_k+1}^{2^{m+1}}(s_{h_k+1}^k),$$

where $h_k$ is the largest index such that $I_h^k = 1$. Note that if $h_k < H$, we have $I_{h_k}^k V_{k,h_k+1}^{2^{m+1}}(s_{h_k+1}^k) \leq 1 = 1 - I_H^k$, and if $h_k = H$, we have $I_{h_k}^k V_{k,h_k+1}^{2^{m+1}}(s_{h_k+1}^k) = 0 = 1 - I_H^k$. Thus, for both cases, we have

$$S_m \leq A_{m+1} + \sum_{k=1}^{K}(1 - I_H^k) + \sum_{k=1}^{K} \sum_{h=1}^{H} I_h^k [V_{k,h}^{2^{m+1}}(s_h^k) - ([\mathbb{P}V_{k,h+1}^{2^m}](s_h^k, a_h^k))^2]. \tag{E.15}$$

For the third term in (E.15), we have

$$\sum_{k=1}^{K} \sum_{h=1}^{H} I_h^k [V_{k,h}^{2^{m+1}}(s_h^k) - ([\mathbb{P}V_{k,h+1}^{2^m}](s_h^k, a_h^k))^2]$$

$$\leq \sum_{k=1}^{K} \sum_{h=1}^{H} I_h^k [V_{k,h}^{2^{m+1}}(s_h^k) - ([\mathbb{P}V_{k,h+1}](s_h^k, a_h^k))^{2^{m+1}}]$$

$$= \sum_{k=1}^{K} \sum_{h=1}^{H} I_h^k (V_{k,h}(s_h^k) - [\mathbb{P}V_{k,h+1}](s_h^k, a_h^k)) \prod_{i=0}^{m}(V_{k,h}^{2^i}(s_h^k) + [\mathbb{P}V_{k,h+1}](s_h^k, a_h^k)^{2^i})$$

$$\leq 2^{m+1} \sum_{k=1}^{K} \sum_{h=1}^{H} I_h^k \max\{V_{k,h}(s_h^k) - [\mathbb{P}V_{k,h+1}](s_h^k, a_h^k), 0\}$$

$$\leq 2^{m+1} \sum_{k=1}^{K} \sum_{h=1}^{H} I_h^k \left[r(s_h^k, a_h^k) + 2\min\{1, \widehat{\beta}_k \left\|\widehat{\mathbf{\Sigma}}_{k,0}^{-1/2} \phi_{k,h,0}\right\|_2\}\right]$$

$$\leq 2^{m+1}(K + 2R_0), \tag{E.16}$$

where the first inequality holds by recursively applying $\mathbb{E}X^2 \geq (\mathbb{E}X)^2$, the second one holds since $V_{k,h} \in [0,1]$, the third one holds due to Lemma E.4, the last one holds due to the definition of $R_0$ (E.7). Substituting (E.16) into (E.15) completes our proof. $\qquad\square$

**Lemma E.7** (Lemma 25, Zhang et al. 2021c). *Let* $\{A_m, S_m\}_{m \in \overline{[M]}}$ *be defined in (E.8) and (E.9). Then we have $\mathbb{P}(\mathcal{E}_{E.7}) > 1 - M\delta$, where*

$$\mathcal{E}_{E.7} := \left\{\forall m \in \overline{[M]}, |A_m| \leq \min\{\sqrt{2\zeta S_m} + \zeta, KH\}\right\}. \tag{E.17}$$

*Proof.* The proof follows the proof of Lemma 25 in Zhang et al. (2021c). We use Lemma F.2 for each $m$. Let $x_{k,h} = I_h^k[[\mathbb{P}V_{k,h+1}^{2^m}](s_h^k, a_h^k) - V_{k,h+1}^{2^m}(s_{h+1}^k)]$, then we have $\mathbb{E}[x_{k,h}|\mathcal{F}_{k,h}] = 0$ and $\mathbb{E}[x_{k,h}^2|\mathcal{F}_{k,h}] = I_h^k[\mathbb{V}V_{k,h+1}^{2^m}](s_h^k, a_h^k)$. Therefore, for each $m \in \overline{[M]}$, with probability at least $1 - \delta$, we have

$$A_m = \sum_{k=1}^{K}\sum_{h=1}^{H} x_{k,h} \leq \sqrt{2\zeta \sum_{k=1}^{K}\sum_{h=1}^{H} I_h^k[\mathbb{V}V_{k,h+1}^{2^m}](s_h^k, a_h^k)} + \zeta.$$

Taking union bound over $m \in \overline{[M]}$ completes the proof. $\square$

**Lemma E.8.** Let $G$ be defined in (E.10). Then we have $G \leq Md\iota/2$.

*Proof.* Recall the definition of $I_H^k$, by the fact that $\det(\widehat{\mathbf{\Sigma}}_{k+1,m}^{-1/2}) < \det(\widetilde{\mathbf{\Sigma}}_{k,H,m}^{-1/2})$, we have

$$(1 - I_H^k) = 1 \Leftrightarrow \exists m \in \overline{[M]}, \; \det(\widehat{\mathbf{\Sigma}}_{k,m}^{-1/2})/\det(\widetilde{\mathbf{\Sigma}}_{k,H,m}^{-1/2}) > 4$$
$$\Rightarrow \exists m \in \overline{[M]}, \; \det(\widehat{\mathbf{\Sigma}}_{k,m}^{-1/2})/\det(\widehat{\mathbf{\Sigma}}_{k+1,m}^{-1/2}) > 4.$$

Let $\mathcal{D}_m$ denote the indices $k$ such that

$$\mathcal{D}_m := \left\{ k \in [K] : \det(\widehat{\mathbf{\Sigma}}_{k+1,m})/\det(\widehat{\mathbf{\Sigma}}_{k,m}) > 16 \right\}.$$

Then we have $G \leq |\cup_{m=0}^{M-1} \mathcal{D}_m| \leq \sum_{m=0}^{M-1} |\mathcal{D}_m|$. For each $m$, we have

$$2|\mathcal{D}_m| < \sum_{k \in \mathcal{D}_m} \log 16 < \sum_{k \in \mathcal{D}_m} \log(\det(\widehat{\mathbf{\Sigma}}_{k+1,m})/\det(\widehat{\mathbf{\Sigma}}_{k,m})) \leq \sum_{k=1}^{K} \log(\det(\widehat{\mathbf{\Sigma}}_{k+1,m})/\det(\widehat{\mathbf{\Sigma}}_{k,m})).$$

Furthermore, by the facts that $\log\det(\widehat{\mathbf{\Sigma}}_{K+1,m}) \leq d\log[\text{tr}(\widehat{\mathbf{\Sigma}}_{K+1,m})/d]$ and $\text{tr}(\widehat{\mathbf{\Sigma}}_{K+1,m}) \leq \text{tr}(\lambda\mathbf{I}) + \sum_{k,h} \|\phi_{k,h,m}\|_2^2/\bar{\sigma}_{k,h,m}^2 \leq d\lambda + KH/\alpha^2$, we have

$$\sum_{k=1}^{K} \log(\det(\widehat{\mathbf{\Sigma}}_{k+1,m})/\det(\widehat{\mathbf{\Sigma}}_{k,m})) = \log(\det(\widehat{\mathbf{\Sigma}}_{K+1,m})/\det(\widehat{\mathbf{\Sigma}}_{1,m}))$$

$$\leq d(\log(\lambda + KH/d\alpha^2) - \log\lambda).$$

Therefore, $|\mathcal{D}_m|$ can be upper bounded by

$$|\mathcal{D}_m| \leq d/2 \cdot \log(1 + KH/(d\lambda\alpha^2)) = d/2 \cdot \iota.$$

Taking the summation over $m$ gives the upper bound of $G$. $\square$

Finally, we define the event $\mathcal{E}_{E.18}$ as

$$\mathcal{E}_{E.18} := \left\{ \sum_{k=1}^{K} \left( \sum_{h=1}^{H} r(s_h^k, a_h^k) - V_1^{\pi^k}(s_1^k) \right) \leq \sqrt{2K\log(1/\delta)} \right\}. \tag{E.18}$$

By Azuma-Hoeffding inequality (Lemma F.1) we have

$$\mathbb{P}(\mathcal{E}_{E.18}) > 1 - \delta. \tag{E.19}$$

With all above lemmas, we are ready to prove Theorem 5.1.

*Proof of Theorem 5.1.* All the following proofs are conditioned on $\mathcal{E}_{E.1} \cap \mathcal{E}_{E.7} \cap \mathcal{E}_{E.18}$, which happens with probability at least $1 - (2M + 1)\delta$ by union bound and the probabilities of individual events $\mathcal{E}_{E.1}, \mathcal{E}_{E.7}, \mathcal{E}_{E.18}$ specified in Lemma E.1, Lemma E.7 and (E.19). First we have $\text{Regret}(K) \leq \sum_{k=1}^{K}[V_{k,1}(s_1^k) - V_1^{\pi^k}(s_1^k)]$ by Lemma E.3. Next we have

$$\sum_{k=1}^{K} V_{k,1}(s_1^k)$$

$$= \sum_{k=1}^{K} \sum_{h=1}^{H} \left[ I_h^k [V_{k,h}(s_h^k) - V_{k,h+1}(s_{h+1}^k)] + (1 - I_h^k)[V_{k,h}(s_h^k) - V_{k,h+1}(s_{h+1}^k)] \right]$$

$$= \sum_{k=1}^{K} \left[ \sum_{h=1}^{H} I_h^k r(s_h^k, a_h^k) + \sum_{h=1}^{H} I_h^k \left[ V_{k,h}(s_h^k) - r(s_h^k, a_h^k) - [\mathbb{P}V_{k,h+1}](s_h^k, a_h^k) \right] \right.$$

$$\left. + \sum_{h=1}^{H} I_h^k [\mathbb{P}V_{k,h+1}](s_h^k, a_h^k) - V_{k,h+1}(s_{h+1}^k) \right] + \sum_{k=1}^{K} \sum_{h=1}^{H} (1 - I_h^k)[V_{k,h}(s_h^k) - V_{k,h+1}(s_{h+1}^k)]$$

$$\leq \sum_{k=1}^{K} \left[ \sum_{h=1}^{H} r(s_h^k, a_h^k) + \sum_{h=1}^{H} I_h^k \left[ V_{k,h}(s_h^k) - r(s_h^k, a_h^k) - [\mathbb{P}V_{k,h+1}](s_h^k, a_h^k) \right] \right.$$

$$\left. + \sum_{h=1}^{H} I_h^k [\mathbb{P}V_{k,h+1}](s_h^k, a_h^k) - V_{k,h+1}(s_{h+1}^k) \right] + \sum_{k=1}^{K} (1 - I_{h_k}^k) V_{k,h_k}(s_{h_k}^k),$$

where $h_k$ is the smallest index such that $I_{h_k}^k = 0$. Then we have

$$\text{Regret}(K)$$

$$\leq \underbrace{\sum_{k=1}^{K} \left( \sum_{h=1}^{H} r(s_h^k, a_h^k) - V_1^{\pi^k}(s_1^k) \right)}_{I_1} + \underbrace{\sum_{k=1}^{K} \sum_{h=1}^{H} I_h^k \left[ V_{k,h}(s_h^k) - r(s_h^k, a_h^k) - [\mathbb{P}V_{k,h+1}](s_h^k, a_h^k) \right]}_{I_2}$$

$$+ \underbrace{\sum_{k=1}^{K} \sum_{h=1}^{H} I_h^k [[\mathbb{P}V_{k,h+1}](s_h^k, a_h^k) - V_{k,h+1}(s_{h+1}^k)]}_{A_0} + \underbrace{\sum_{k=1}^{K} (1 - I_H^k)}_{G}$$

$$\leq \sqrt{2K \log(1/\delta)} + 2R_0 + |A_0| + G. \tag{E.20}$$

The bound of $I_1$ comes from $\mathcal{E}_{E.18}$. For $I_2$, by Lemma E.4 we have

$$I_2 \leq 2 \sum_{k=1}^{K} \sum_{h=1}^{H} I_h^k \min\{1, \widehat{\beta}_k \left\| \widehat{\mathbf{\Sigma}}_{k,0}^{-1/2} \phi_{k,h,0} \right\|_2 \} = 2R_0.$$

Next we bound $2R_0 + |A_0|$ in (E.20). Substituting (E.14) in Lemma E.6 into (E.17) in Lemma E.7, we have

$$|A_m| \leq \sqrt{2\zeta(|A_{m+1}| + G + 2^{m+1}(K + 2R_0))} + \zeta$$
$$\leq \sqrt{2\zeta}\sqrt{|A_{m+1}| + 2^{m+1}(K + 2R_0)} + \sqrt{2\zeta G} + \zeta \tag{E.21}$$

Substituting (E.14) in Lemma E.6 into (E.11) in Lemma E.5, we have

$$R_m \leq 4d\iota + 4\widehat{\beta}_K \gamma^2 d\iota + 2\widehat{\beta}_K \sqrt{d\iota}\sqrt{|A_{m+1}| + G + 2^{m+1}(K + 2R_0) + 4R_m + 2R_{m+1} + KH\alpha^2}$$
$$\leq 2\widehat{\beta}_K \sqrt{d\iota}\sqrt{|A_{m+1}| + 2^{m+1}(K + 2R_0) + 4R_m + 2R_{m+1}} +$$
$$+ \underbrace{4d\iota + 4\widehat{\beta}_K \gamma^2 d\iota + 2\widehat{\beta}_K \sqrt{d\iota}\sqrt{G + KH\alpha^2}}_{I_c} \tag{E.22}$$

Calculating (E.21) + 2×(E.22) and using $\sqrt{a} + \sqrt{b} \leq \sqrt{2(a + b)}$, we have

$$|A_m| + 2R_m$$
$$\leq 2I_c + \sqrt{2\zeta G} + \zeta + \sqrt{2} \max\{4\widehat{\beta}_K \sqrt{d\iota}, \sqrt{2\zeta}\}$$
$$\cdot \sqrt{|A_{m+1}| + 2^{m+1}(K + 2R_0) + 4R_m + 2R_{m+1} + |A_{m+1}| + 2^{m+1}(K + 2R_0)}$$
$$\leq 2I_c + \sqrt{2\zeta G} + \zeta + 2 \max\{4\widehat{\beta}_K \sqrt{d\iota}, \sqrt{2\zeta}\}$$
$$\cdot \sqrt{|A_m| + 2R_m + |A_{m+1}| + 2R_{m+1} + 2^{m+1}(K + |A_0| + 2R_0)}$$

Then by Lemma F.7 with $a_m = |A_m| + 2R_m \le 3KH$ and $M = \log(3KH)/\log 2$, $|A_0| + 2R_0$ can be bounded as

$$
\begin{aligned}
&|A_0| + 2R_0 \\
&\le 22 \cdot 8 \max\{2\widehat{\beta}_K^2 d\iota, \zeta\} + 6 \cdot (2I_c + \sqrt{2\zeta G} + \zeta) + 8 \max\{4\widehat{\beta}_K \sqrt{d\iota}, \sqrt{2\zeta}\} \sqrt{2(K + |A_0| + 2R_0)} \\
&\le 176 \max\{2\widehat{\beta}_K^2 d\iota, \zeta\} + 6(8d\iota + 8\widehat{\beta}_K \gamma^2 d\iota + 4\widehat{\beta}_K \sqrt{d\iota}\sqrt{G + KH\alpha^2} + \sqrt{2\zeta G} + \zeta) \\
&\quad + 16 \max\{2\widehat{\beta}_K \sqrt{2d\iota}, \sqrt{\zeta}\}\sqrt{K} + 16 \max\{2\widehat{\beta}_K \sqrt{2d\iota}, \sqrt{\zeta}\}\sqrt{|A_0| + 2R_0}. \quad\quad \text{(E.23)}
\end{aligned}
$$

By the fact $x \le a\sqrt{x} + b \Rightarrow x \le 2a^2 + 2b$, (E.23) implies

$$
\begin{aligned}
|A_0| + 2R_0 &\le 864 \max\{8\widehat{\beta}_K^2 d\iota, \zeta\} + 12(8d\iota + 8\widehat{\beta}_K \gamma^2 d\iota + 4\widehat{\beta}_K \sqrt{d\iota}\sqrt{G + KH\alpha^2} + \sqrt{2\zeta G} + \zeta) \\
&\quad + 32 \max\{2\widehat{\beta}_K \sqrt{2d\iota}, \sqrt{\zeta}\}\sqrt{K}. \quad\quad \text{(E.24)}
\end{aligned}
$$

Finally, substituting (E.24) into (E.20) and bounding $G$ by Lemma E.8, the regret is bounded as

$$
\begin{aligned}
\text{Regret}(K) &\le 12(8d\iota + 8\widehat{\beta}_K \gamma^2 d\iota + 4\widehat{\beta}_K \sqrt{d\iota}\sqrt{Md\iota/2 + KH\alpha^2} + \sqrt{Md\iota\zeta} + \zeta) \\
&\quad + 864 \max\{8\widehat{\beta}_K^2 d\iota, \zeta\} + Md\iota/2 + \left[\sqrt{2\log(1/\delta)} + 32 \max\{2\widehat{\beta}_K \sqrt{2d\iota}, \sqrt{\zeta}\}\right]\sqrt{K},
\end{aligned}
$$

which completes our proof. $\square$

## E.3 Proof of Theorem 5.3

We have the following lemma to lower bound the regret for linear bandits.

**Lemma E.9** (Lemma 25, Zhou et al. 2021a). Fix a positive real $0 < \delta \le 1/3$, and positive integers $K, d$ and assume that $K \ge d^2/(2\delta)$. Let $\Delta = \sqrt{\delta/K}/(4\sqrt{2})$ and consider the linear bandit problems $\mathcal{L}_{\boldsymbol{\mu}}$ parameterized with a parameter vector $\boldsymbol{\mu} \in \{-\Delta, \Delta\}^d$ and action set $\mathcal{A} = \{-1, 1\}^d$ so that the reward distribution for taking action $\boldsymbol{a} \in \mathcal{A}$ is a Bernoulli distribution $B(\delta + \langle \boldsymbol{\mu}^*, \boldsymbol{a} \rangle)$. Then for any bandit algorithm $\mathcal{B}$, there exists a $\boldsymbol{\mu}^* \in \{-\Delta, \Delta\}^d$ such that the expected pseudo-regret of $\mathcal{B}$ over first $K$ steps on bandit $\mathcal{L}_{\boldsymbol{\mu}^*}$ is lower bounded as follows:

$$
\mathbb{E}_{\boldsymbol{\mu}^*}\text{Regret}(K) \ge \frac{d\sqrt{K\delta}}{8\sqrt{2}}.
$$

*Proof of Theorem 5.3.* The linear mixture MDP instance is similar to the MDP instances considered in Zhou et al. (2021b,a); Zhang et al. (2021a). The state space $\mathcal{S} = \{x_1, x_2, x_3\}$. The action space $\mathcal{A} = \{\mathbf{a}\} = \{-1, +1\}^{d-1}$. The reward function satisfies $r(x_1, \mathbf{a}) = r(x_2, \mathbf{a}) = 0$ and $r(x_3, \mathbf{a}) = 1/H$. The transition probability satisfies $\mathbb{P}(x_2|x_1, \mathbf{a}) = 1 - (\delta + \langle \boldsymbol{\mu}, \mathbf{a} \rangle)$ and $\mathbb{P}(x_3|x_1, \mathbf{a}) = \delta + \langle \boldsymbol{\mu}, \mathbf{a} \rangle$, where $\delta = 1/6$ and $\boldsymbol{\mu} \in \{-\Delta, \Delta\}^{d-1}$ with $\Delta = \sqrt{\delta/K}/(4\sqrt{2})$.

First, similar to the proof in (Section E.1, Zhou et al. 2021a), we can verify that when $K \ge (d-1)/(192(B-1))$, our instance is a $B$-bounded linear mixture MDP with $\mathbb{P}(s'|s, \mathbf{a}) = \langle \boldsymbol{\phi}(s'|s, \mathbf{a}), \boldsymbol{\theta} \rangle$, where

$$
\boldsymbol{\phi}(s'|s, \mathbf{a}) = \begin{cases}
(\alpha(1-\delta), -\beta\mathbf{a}^\top)^\top, & s = x_1, s' = x_2; \\
(\alpha\delta, \beta\mathbf{a}^\top)^\top, & s = x_1, s' = x_3; \\
(\alpha, \mathbf{0}^\top)^\top, & s \in \{x_2, x_3\}, s' = s; \\
\mathbf{0}, & \text{otherwise}.
\end{cases}
$$
$$
\boldsymbol{\theta} = (1/\alpha, \boldsymbol{\mu}^\top/\beta)^\top.
$$

where $\alpha = \sqrt{1/(1 + \Delta(d-1))}$, $\beta = \sqrt{\Delta/(1 + \Delta(d-1))}$.

Second, our instance can be regarded as a linear bandit instance with a Bernoulli reward distribution $B(\delta + \langle \boldsymbol{\theta}, \mathbf{a} \rangle)$. Therefore, the lower bound of regret for linear mixture MDP directly follows the regret for linear bandits in Lemma E.9, by picking $\boldsymbol{\mu} = \boldsymbol{\mu}^*$ and $\delta = 1/6$. $\square$

# F  Auxiliary lemmas

**Lemma F.1** (Azuma-Hoeffding inequality, Azuma 1967). Let $M > 0$ be a constant. Let $\{x_i\}_{i=1}^n$ be a stochastic process, $\mathcal{G}_i = \sigma(x_1, \ldots, x_i)$ be the $\sigma$-algebra of $x_1, \ldots, x_i$. Suppose $\mathbb{E}[x_i|\mathcal{G}_{i-1}] = 0$, $|x_i| \le M$ almost surely. Then, for any $0 < \delta < 1$, we have

$$\mathbb{P}\left( \sum_{i=1}^n x_i \le M\sqrt{2n \log(1/\delta)} \right) > 1 - \delta.$$

**Lemma F.2** (Lemma 11, Zhang et al. 2021d). Let $M > 0$ be a constant. Let $\{x_i\}_{i=1}^n$ be a stochastic process, $\mathcal{G}_i = \sigma(x_1, \ldots, x_i)$ be the $\sigma$-algebra of $x_1, \ldots, x_i$. Suppose $\mathbb{E}[x_i|\mathcal{G}_{i-1}] = 0$, $|x_i| \le M$ and $\mathbb{E}[x_i^2|\mathcal{G}_{i-1}] < \infty$ almost surely. Then, for any $\delta, \epsilon > 0$, we have

$$\mathbb{P}\left( \left| \sum_{i=1}^n x_i \right| \le 2\sqrt{2\log(1/\delta) \sum_{i=1}^n \mathbb{E}[x_i^2|\mathcal{G}_{i-1}]} + 2\sqrt{\log(1/\delta)}\epsilon + 2M\log(1/\delta) \right)$$
$$> 1 - 2(\log(M^2 n/\epsilon^2) + 1)\delta.$$

**Lemma F.3** (Unbounded Freedman's inequality, Dzhaparidze and Van Zanten 2001; Fan et al. 2017). Let $\{x_i\}_{i=1}^n$ be a stochastic process, $\mathcal{G}_i = \sigma(x_1, \ldots, x_i)$ be the $\sigma$-algebra of $x_1, \ldots, x_i$. Suppose $\mathbb{E}[x_i|\mathcal{G}_{i-1}] = 0$ and $\mathbb{E}[x_i^2|\mathcal{G}_{i-1}] < \infty$ almost surely. Then, for any $a, v, y > 0$, we have

$$\mathbb{P}\left( \sum_{i=1}^n x_i > a, \ \sum_{i=1}^n \left( \mathbb{E}[x_i^2|\mathcal{G}_{i-1}] + x_i^2 \mathbb{1}\{|x_i| > y\} \right) < v^2 \right) \le \exp\left( \frac{-a^2}{2(v^2 + ay/3)} \right).$$

**Lemma F.4** (Lemma 8, Zhang et al. 2021c). Let $\{x_i \ge 0\}_{i \ge 1}$ be a stochastic process, $\{\mathcal{G}_i\}_{i \ge 1}$ be a filtration satisfying $x_i$ is $\mathcal{G}_i$-measurable. We also have $|x_i| \le 1$. Then for any $c \ge 1$ we have

$$\mathbb{P}\left( \exists n, \ \sum_{i=1}^n x_i \ge 4c\log(4/\delta), \ \sum_{i=1}^n \mathbb{E}[x_i|\mathcal{G}_{i-1}] \le c\log(4/\delta) \right) \le \delta.$$

**Lemma F.5** (Lemma 11, Abbasi-Yadkori et al. 2011). For any $\lambda > 0$ and sequence $\{\mathbf{x}_k\}_{k=1}^K \subset \mathbb{R}^d$ for $k \in [K]$, define $\mathbf{Z}_k = \lambda\mathbf{I} + \sum_{i=1}^{k-1} \mathbf{x}_i\mathbf{x}_i^\top$. Then, provided that $\|\mathbf{x}_k\|_2 \le L$ holds for all $k \in [K]$, we have

$$\sum_{k=1}^K \min\{1, \|\mathbf{x}_k\|_{\mathbf{Z}_k^{-1}}^2\} \le 2d\log(1 + KL^2/(d\lambda)).$$

**Lemma F.6** (Lemma 12, Abbasi-Yadkori et al. 2011). Suppose $\mathbf{A}, \mathbf{B} \in \mathbb{R}^{d \times d}$ are two positive definite matrices satisfying $\mathbf{A} \succeq \mathbf{B}$, then for any $\mathbf{x} \in \mathbb{R}^d$, $\|\mathbf{x}\|_{\mathbf{A}} \le \|\mathbf{x}\|_{\mathbf{B}} \cdot \sqrt{\det(\mathbf{A})/\det(\mathbf{B})}$.

**Lemma F.7** (Lemma 12, Zhang et al. 2021c). Let $\lambda_1, \lambda_2, \lambda_4 > 0$, $\lambda_3 \ge 1$ and $\kappa = \max\{\log_2 \lambda_1, 1\}$. Let $a_1, \ldots, a_\kappa$ be non-negative real numbers such that $a_i \le \min\{\lambda_1, \lambda_2\sqrt{a_i + a_{i+1} + 2^{i+1}\lambda_3} + \lambda_4\}$ for any $1 \le i \le \kappa$. Let $a_{\kappa+1} = \lambda_1$. Then we have $a_1 \le 22\lambda_2^2 + 6\lambda_4 + 4\lambda_2\sqrt{2\lambda_3}$.