# OpenReview forum: "Computationally Efficient Horizon-Free Reinforcement Learning for Linear Mixture MDPs"
_NeurIPS.cc/2022/Conference — NeurIPS 2022 Accept_

### Official Review · Reviewer_XjDu · 2022-06-19

**Rating:** 5
**Confidence:** 4
**Soundness:** 3 good
**Presentation:** 2 fair
**Contribution:** 2 fair

**Summary:**

This paper proposed a computationally efficient horizon-free RL algorithm.

**Questions:**

1. If the cumulative reward is bounded by H, will the bound still be independent of H?

**Ethics Review Area:**

["I don’t know"]

**Strengths And Weaknesses:**

The writing needs improvements and the notation is a bit heavy. The contribution is not very significant. UCRL-VTR itself is not computationally-efficient beyond linear case, as pointet out in "Model-Based Reinforcement Learning with Value-Targeted
Regression".

I feel the algorithm used in this paper does not show significant improvement over the one in "Nearly minimax optimal reinforcement learning for linear mixture markov decision processes." The weighted OFUL+UCRL VTR is almost the same. The improvement of regret bound for linear bandit is not huge since dsqrt{sum sigma_i^2} term is the leading term.

The main trick to obtain horizon-free bound seems to be cumulative rewards bounding by [0, 1] + variance-aware confidence interval. This is known before and this paper seems to offer no new message in my mind.

I think Algorithm 2 heavily relies on linear mixture MDP assumption. That means you assume the complicated environment enjoys a quite simplified parametric form. If your algorithm is general but you use linear mixture MDP as a sanity check, that's fine. But the algorithm now is designed for linear mixture MDPs.

Since the paper claims the major contribution is computational-wise, I feel some simple experiments are needed. It is unclear what's the empirical performance since it is not built on any algorithm that is widely empirically evaluated. It will be good to observe what's the meaning of horizon free on existing benchmarks.

---

> ### Author Response · Authors · 2022-08-02
> **To Reviewer XjDu**
>
> Thanks for your positive comments!
>
> **Q1** The contribution is not very significant. UCRL-VTR itself is not computationally-efficient beyond linear case.
>
> **A1** We respectfully disagree with the reviewer’s comment that our contribution is not significant. Our work indeed focuses on the linear function approximation rather than general function approximation. However, even for the simplest possible linear function approximation case, how to achieve horizon-free regret efficiently for linear mixture MDPs remains an open problem. Our work has improved all existing horizon-free algorithms for linear mixture MDPs (Zhang et al., 2021c; Kim et al., 2021).
> ***
> **Q2** No big improvement from Zhou et al. (2021). Improvement of linear bandit regret is not huge since the leading term is $d\sqrt{\sum_{k=1}^K \sigma_k^2}$.
>
> **A2** The reviewer’s comment is incorrect. We would like to point out that the ‘leading terms’ of the linear bandit regret of WeightedOFUL in Zhou et al. (2021) should be $\tilde O(d\sqrt{\sum_{k=1}^K \sigma_k^2} + \sqrt{dK})$, and we cannot omit $\sqrt{dK}$ without making additional assumptions. In particular, for the case where $d\sqrt{\sum_{k=1}^K \sigma_k^2}$ is small, the summation of variances is bounded by a constant, then the leading term of WeightedOFUL will be $\tilde O(\sqrt{dK})$. In sharp contrast, the regret of WeightedOFUL+ only has the term $\tilde O(d\sqrt{\sum_{k=1}^K \sigma_k^2}$, which strictly improves WeightedOFUL by a large margin.
> ***
> **Q3** To obtain a horizon-free regret, bounded value assumption and variance-aware confidence sets are required. Neither of them is new.
>
> **A3** Bounded value assumption is standard and needed in all existing horizon-free papers, In addition, simply having bounded value assumption and variance-aware confidence sets is not sufficient to achieve horizon-free regret. In detail, bounded rewards assumption and variance-aware confidence sets only yields a regret in the form of $\tilde O(\sqrt{d^2 +dH}\sqrt{K} + d^2H^2 + d^3H)$, as suggested by Remark 22 in Zhou et al. (2021). You can see that there is still polynomial dependence on H (horizon) in the non-dominating terms, and this is not considered as a horizon-free regret. To further eliminate the poly H dependence, we need to use the high-order momentum estimator (HOME) in Algorithm 3, which is a new technique proposed in our paper.
> ***
> **Q4** Algorithm 2 relies on linear mixture MDP model.
>
> **A4** In this work, we start from linear mixture MDPs for the following reasons. First, linear mixture MDPs and linear MDPs are two main nontabular parameteric MDP classes studied by RL researchers. Thus, any improvements over linear mixture MDPs can greatly aid the following research for nontabular MDPs. Second, although linear mixture MDPs are not the most general MDP classes studied by RL researchers, our proposed algorithms, with the horizon-free regret, will definitely help people to design more efficient algorithms for general MDPs using similar ideas. Therefore, we take linear mixture MDPs as a starting point.
> ***
> **Q5** Experiments
>
> **A5** We have added experiments to Appendix A in the revised paper. We adopt the RiverSwim environment introduced by Strehl and Littman (2008) and considered by Ayoub et al. (2020). We compare our HF-UCRL-VTR+ with several baseline algorithms including UCRL-VTR+, UCRL-VTR, Q-learning and the randomly action selection. From the experimental results, we can draw the following conclusions: (1) HF-UCRL-VTR$^+$ with $M = 4$ outperforms HF-UCRL-VTR$^+$ with $M = 2$, which validates the advantage of the HOME estimator. (2) HF-UCRL-VTR$^+$ and UCRL-VTR$^+$ outperforms UCRL-VTR, which suggests that the Bernstein-type bonus is useful. (3) All UCRL-VTR-based algorithms outperform the tabular Q-learning, The average reward of UCRL-VTR-based algorithms is generally higher than that of Q-learning, This suggests the validity of the value-targeted regression, as well as the use of the feature $\phi$.
> ***
> **Q6** If the cumulative reward is bounded by H, will the bound still be independent of H?
>
> **A6** In general, the answer is ‘no’. Take the hard instance considered in Theorem 5.3 for example. We set the reward at state $x_3$ as $r(x_3, a) = 1$. Then the revised MDP is a linear mixture MDP with a $H$-bounded cumulative reward, but any algorithm will suffer an $\Omega(dH\sqrt{K})$ regret according to our proof of Theorem 5.3.

---

### Official Review · Reviewer_xqq7 · 2022-06-27

**Rating:** 8
**Confidence:** 4
**Soundness:** 3 good
**Presentation:** 3 good
**Contribution:** 4 excellent

**Summary:**

This paper gives the first computationally efficient horizon-free algorithm for linear mixture MDPs, which achieves the optimal $O(d\sqrt{K} + d^2)$ regret up to logarithmic factors. Their technique can also be applied to heterogeneous linear bandits and achieve $d\sqrt{\sum_{k=1}^K\sigma_k^2} + d$ regret when the reward variance $\sigma_k^2$ is known.

**Questions:**

Can this technique be applied to linear MDP and achieve horizon-free regret?

**Limitations:**

Nothing necessary stands out.

**Strengths And Weaknesses:**

This paper solves a very important open problem in RL with linear function approximation: getting horizon-free regret bound efficiently. The developed techniques can potentially be applied to many other problems and improved current results. The algorithm also has a simple structure and thus easy to implement and extend. Discussions on related work and implications of results are thorough and well presented.

Given that the algorithm is simple enough, it would be beneficial to also perform experiments to verify the theoretical gain.

---

> ### Author Response · Authors · 2022-08-02
> **To Reviewer xqq7**
>
> Thanks for recognizing the significance and contribution of our work!
>
> **Q1** Experiments.
>
> **A1** We have added experiments to Appendix A in the revised paper. We adopt the RiverSwim environment introduced by Strehl and Littman (2008) and considered by Ayoub et al. (2020). We compare our HF-UCRL-VTR+ with several baseline algorithms including UCRL-VTR+, UCRL-VTR, Q-learning and the randomly action selection. From the experimental results, we can draw the following conclusions: (1) HF-UCRL-VTR$^+$ with $M = 4$ outperforms HF-UCRL-VTR$^+$ with $M = 2$, which validates the advantage of the HOME estimator. (2) HF-UCRL-VTR$^+$ and UCRL-VTR$^+$ outperforms UCRL-VTR, which suggests that the Bernstein-type bonus is useful. (3) All UCRL-VTR-based algorithms outperform the tabular Q-learning, The average reward of UCRL-VTR-based algorithms is generally higher than that of Q-learning, This suggests the validity of the value-targeted regression, as well as the use of the feature $\phi$.
> ***
> **Q2** Can we achieve horizon-free regret on linear MDPs?
>
> **A2** We are not sure about that, since algorithms for linear MDPs often need to deal with the covering number of the value function class, which may introduce additional and (potentially) unavoidable terms that depend on $H$. We will investigate it in our future work.

---

### Official Review · Reviewer_yBHw · 2022-07-07

**Rating:** 7
**Confidence:** 4
**Soundness:** 4 excellent
**Presentation:** 3 good
**Contribution:** 3 good

**Summary:**

This paper extends the previous work [1], by proposing a novel variance-aware and uncertainty-aware weight in the weighted least square regression to provide an improved analysis on the heterogeneous linear bandits and linear mixture MDPs, that can be computational efficient for linear mixture MDPs and for the case when we have an upper bound of the noise scale in heterogeneous linear bandits. The key observation is that, by introducing the additional uncertainty aware terms in the weight, we can have much tighter confidence set when $\sigma_k$ is small and $K$ is large, as illustrated in Lemma 4.3 and Line 213-216.

[1] Zhou, Dongruo, Quanquan Gu, and Csaba Szepesvari. "Nearly minimax optimal reinforcement learning for linear mixture markov decision processes." Conference on Learning Theory. PMLR, 2021.


**Questions:**

* This paper is in general good. Mainly I hope the authors provide more intuition and discussion on Lemma 4.3, especially how it compares with the self-normalized martingale concentration inequality in [1]. It may not be so simple for the potential audience to understand the motivation.

* Additional Minor Issues:
* * Please check the input of the algorithm. Some hyperparameters should appear in the input part (e.g. $\alpha$ and $\gamma$ in Algorithm 1).
* * It would be better if the authors can make some additional discussions on the work without the assumption of known variance. As far as I know, unlike the authors claim in Line 84-87, both Zhang et al. and Kim et al. do not require an upper bound of $\sigma_k^2$. And to deal with this issue, they need to work on the noise concentration and use some peeling techniques to construct the confidence set. I hope the authors make this distinction clear.

[1] Zhou, Dongruo, Quanquan Gu, and Csaba Szepesvari. "Nearly minimax optimal reinforcement learning for linear mixture markov decision processes." Conference on Learning Theory. PMLR, 2021.


**Limitations:**

This paper is on the theory side and does not need to address the societal impact.

**Strengths And Weaknesses:**

### Strengths:
* Simple but clear theoretical results.
* Well-written in general.

### Weaknesses:
* The authors do not provide so much intuition on the key results (Lemma 4.3). I think it deserves much more discussion.
* A minor concern: the part other than Lemma 4.3 is not technically significant.

---

> ### Author Response · Authors · 2022-08-02
> **To Reviewer yBHw**
>
> Thanks for your positive comments!
>
> **Q1** Intuition and discussion about Lemma 4.3.
>
> **A1** We compare Lemma 4.3 and its counterpart, the self-normalized martingale Bernstein inequality in Zhou et al. (2021). The confidence radius $\beta_k$ in Lemma 4.3 is of order
> $\tilde O(\sigma \sqrt{d} + \max_{1 \leq i \leq k}|\eta_i| \min(1, w_i) )$ with a small enough $\epsilon$, while the $\beta_k$ in Zhou et al. (2021) is of order $\tilde O(\sigma\sqrt{d} + R)$. Lemma 4.3 strictly improves Zhou et al. (2021) since $|\eta_i| \leq R$. The intuition why we introduce the improved Bernstein inequality in Lemma 4.3 is that it suggests the confidence radius $\beta_k$ only depends on an adaptive quantity $\max_{1 \leq i \leq k}|\eta_i| \min(1, w_i) )$, rather a worst-case constant $R$. Thus, combining the uncertainty-aware weight $\sigma_k$ defined in (4.1), we are able to obtain an improved regret as we have demonstrated from line 224 to line 230.
> ***
> **Q2** The part other than Lemma 4.3 is not technically significant.
>
> **A2** We believe the reviewer has overlooked our other contributions. In fact, with only Lemma 4.3, we are not able to improve the regret for linear bandits in Zhou et al. (2021). The algorithm modification (uncertainty-aware weight $\sigma_k$) and the improved elliptical lemma (Lemma 4.4) are very important as well. Moreover, for linear mixture MDPs, the HOME estimator (Algorithm 3) is new and it has never been proposed before. It plays an important role for obtaining a horizon-free regret.
>
> ***
>
> **Q3** Input of the algorithm.
>
> **A3** We have revised the description of Algorithm 1 to add the input hyperparameters.
> ***
> **Q4** More discussion about Zhang et al. and Kim et al. in related work.
>
> **A4** Both Zhang et al. (2021c) and Kim et al. (2021) do not need to know the variance information in their algorithm. Instead, they construct their confidence sets based on the peeling technique directly on the noise itself, which leads to a final regret bound based on the variances of the noises. However, as we commented in our paper, neither of their algorithms is computationally efficient. We will add more discussion on these two works in the revision.

---

### Official Review · Reviewer_Z3Fm · 2022-07-11

**Rating:** 7
**Confidence:** 2
**Soundness:** 3 good
**Presentation:** 3 good
**Contribution:** 3 good

**Summary:**

This paper studies the problem of Episodic Linear Mixture MDPs, where the transition model is a linear combination of $d$ valid MDP transition models. The MDP has a countable set of states and a finite set of actions, and the cumulative reward in each episode is not larger than $1$. The agent interacts with the MDP for $K$ epsidoes, and the planning horizon is $H$ for each episode. As a warm-up, this paper gives an algorithm WeightedOFUL$^+$ for heterogeneous linear bandits. It is computationally efficient and achieves a state-of-the-art $\tilde{O}(d \sqrt{\sum_{k=1}^K \sigma_k^2} + d)$ regret, where $\sigma_k$ is the variance of noise in episode $k$ (assumed to be known). The main result is a computationally efficient algorithm, HF-UCRL-VTR$^+$, for Linear Mixture MDPs. It achieves a $\tilde{O}(d \sqrt{K} + d)$ regret, matching the lower-bound given in this paper.

**Questions:**

None.

**Limitations:**

None.

**Strengths And Weaknesses:**

Strengths:
1. This paper improves upon previous results of Linear MABs and Linear Mixture MDPs by giving computational efficient algorithms with matching state-of-the-art regrets. For Linear Mixture MDP, this paper also provides a tight lower-bound.
2. This paper is clearly written. The comparison between different works is straightforward.
3. This work is significant because it nearly closes the problem of Linear Mixture MDP.


Weaknesses:
1. WeightedOFUL$^+$ needs knowledge of $\sigma_k$, which is in general not practical, otherwise, it would degenerate into $d\sqrt{K}$.

---

> ### Author Response · Authors · 2022-08-02
> **To Reviewer Z3Fm**
>
> Thanks for your positive comments!
>
> **Q1** WeightedOFUL needs knowledge of variance, which is in general not practical.
>
> **A1** It is worth noting that WeightedOFUL actually only requires the upper bound of the underlying variances. Therefore, WeightedOFUL is still practical for the problem where the variance upper bounds can be estimated, for example, linear mixture MDP as we have introduced in Section 5. To completely avoid any knowledge of the variances, we may utilize the implicit confidence set introduced by Zhang et al. (2021c), Kim et al. (2021), which makes their algorithms computationally intractable. We leave it as future work to propose a computationally efficient algorithm without the knowledge of variance for the linear contextual bandits.

---

> > ### Comment · Reviewer_Z3Fm · 2022-08-05
> > **post rebuttal response**
> >
> > After read the response, I decide to keep the score unchanged.

---

> > > ### Author Response · Authors · 2022-08-06
> > > **Re: post rebuttal response**
> > >
> > > Thank you for your positive feedback!

---

### Meta-Review · Area_Chair_m3Hk · 2022-08-26

**Recommendation:** Accept
**Confidence:** Certain

**Metareview:**

This work advances the state-of-the-art for horizon-free regret bounds for linear mixture MDP as well as heterogeneous linear bandits. Clear accept.

**Award:**

No

---

### Decision · Program_Chairs · 2022-09-14

Accept